



**Current Understanding of the Driving Mechanisms for Spatiotemporal**
**Variations of Atmospheric Speciated Mercury: A Critical Review**
Huiting Mao[1*], Irene Cheng[2], and Leiming Zhang[2]
[1]Department of Chemistry, State University of New York College of Environmental Science and
Forestry, Syracuse, NY 13210
[2]Air Quality Research Division, Science and Technology Branch, Environment and Climate
Change Canada, 4905 Dufferin Street, Toronto, Ontario, M3H 5T4, Canada
[*]Corresponding author: hmao@esf.edu



**Abstract**
16          Understanding of spatial and temporal variations of atmospheric speciated mercury can

advance our knowledge of mercury cycling in various environments.  This review summarized
spatiotemporal variations of total gaseous mercury or gaseous elemental mercury (TGM/GEM),
gaseous oxidized mercury (GOM), and particulate-bound mercury (PBM) in various
environments including oceans, continents, high elevation, the free troposphere, and low to high
latitudes. In the marine boundary layer (MBL), the oxidation of GEM was generally thought to
drive the diurnal and seasonal variations of TGM/GEM and GOM in most oceanic regions,
leading to lower GEM and higher GOM from noon to afternoon and higher GEM during winter
and higher GOM during spring-summer.  At continental sites, the driving mechanisms of
TGM/GEM diurnal patterns included surface and local emissions, boundary layer dynamics,
GEM oxidation, and mountain-valley winds at high elevation sites.  Oxidation of GEM and
entrainment of GOM from the free troposphere influenced the diurnal patterns of GOM at
continental sites.  No pronounced diurnal variation was found for Tekran measured PBM at
MBL and continental sites.  Seasonal variations in TGM/GEM at continental sites were
attributed to increased winter combustion, increased surface emissions during summer, and
monsoons in Asia.  GEM oxidation, free tropospheric transport, anthropogenic emissions, and
wet deposition appeared to affect the seasonal pattern of GOM at continental sites.  Since
measurements were predominantly in the northern hemisphere (NH), increased PBM at
continental sites during winter was primarily due to local/regional coal combustion and wood
burning emissions.  Long-term TGM measurements from the MBL and continental sites
indicated an overall declining trend consistent with those of anthropogenic and natural emissions
and potentially redox chemistry.  The latitudinal gradient in TGM/GEM showed an increase from



the southern to northern hemisphere due largely to the vast majority of Hg emissions in the NH.
This gradient was insignificant during summer probably as a result of stronger meridional
mixing.  Aircraft measurements indicated no significant GEM gradient with altitude over the
field campaign regions; however depletion of GEM was observed in air masses under
stratospheric influence. Remaining questions and issues related to factors potentially contributing
to the observed spatiotemporal variations were identified, and recommendations for future
research needs were provided.
**1. Introduction**

Atmospheric mercury (Hg) is a pervasive toxic with comparable natural and

anthropogenic sources (UNEP, 2013).  It is operationally defined in three forms, gaseous
elemental mercury (GEM), gaseous oxidized mercury (GOM), and particulate-bound mercury
(PBM).  In most environments GEM comprises >95% of total gaseous mercury (TGM =
GEM+GOM) with lifetime of 0.5 – 1 year (Driscoll et al., 2013).  Besides emissions, GOM and
PBM are largely formed from oxidation of GEM, with lifetimes of hours to weeks (Cole et al.,
2014).  They are highly soluble, and their wet and dry deposition is a major input of Hg to
ecosystems and oceans followed by bioaccumulation, where Hg can enter human bodies through
the food chain.  To ultimately regulate anthropogenic emissions of Hg in order to control the
ambient atmospheric concentration of Hg, it is imperative to understand Hg cycling between the
atmosphere, ecosystems, and oceans.

The pathways of Hg cycling include chemical transformation and transport via air and

water in various systems as illustrated in Subir et al. (2011).  Mercury can be chemically



transformed from one species to another through oxidation/reduction reactions, complex
formation, phase transitions, biodegradation, and surface and heterogeneous interactions with
aerosols, clouds, snow, and ice.  Mercury can also be redistributed between geographic locations
and spheres through physical processes such as wind, water runoff, dry and wet deposition, and
volatilization.  In addition, natural and anthropogenic sources of Hg are distributed vastly uneven
as a result of anthropogenic activities and land surface types.  The eventual effect of all these
processes, some of which are in fact sinks, and sources is manifested in the great heterogeneity
of temporal and spatial variations of atmospheric Hg concentrations observed in numerous
studies (Sprovieri et al, 2010b, references therein; references in Tables S1 – S7 in the
supplementary information (SI)).  Characterization and intercomparison of such variations for
different geographic and chemical environments can provide a gateway to our understanding of
Hg cycling.

Numerous measurement studies in the literature have shown distinctly different

spatiotemporal variations of GEM, GOM, and PBM in the following environments:
• Marine boundary layer
• Land: urban, rural, and remote
• High elevation, high altitude
• Low, mid-, and high latitudes
owing to their respective atmospheric chemical composition, sources, and meteorological
conditions. In spite of our nebulous understanding of chemical transformation of atmospheric Hg,
it is commonly thought that GEM is oxidized by halogen radicals (e.g., Br, BrO, ClO, BrCl),
ozone ($O_3$), and hydroxyl radicals (OH) in gas and/or solid phase (Hynes et al., 2009, references
therein). A more recent quantum calculation study suggested that more abundant radicals such as



NO, NO$_2$, HO$_2$, ClO, or BrO could more readily oxidize GEM (Dibble et al., 2012). In
springtime Antarctic and Arctic regions, where there were relatively more abundant halogen
radicals, it was observed that GEM was depleted to very low levels accompanied by hundreds of
picograms GOM (e.g., Schroeder and Munthe, 1998; Lindberg et al., 2002). The diurnal and
seasonal variation of GEM, GOM, and PBM appeared to be highly correlated with that of BrO
leading to the hypothesis of GEM oxidation by Br and BrO. In the marine boundary layer
(MBL) over the Dead Sea and Cape point, South Africa, similar GEM depletion was also
observed (Brunke et al., 2010; Obrist et al., 2011), which was hypothesized to be associated with
GEM oxidation by bromine-related species (Obrist et al., 2011). In most marine environments,
however, GEM depletion events have not been observed.

Over land, spatiotemporal variations of GEM, GOM, and PBM exhibited different

characteristics from over the ocean. Also, they appeared to differ greatly from urban to remote
areas, from the surface to the free troposphere, from low to high latitudes, from the northern to
southern hemisphere, and between different geographic locations of the same environment type.
For example, GEM concentrations in urban locations were often observed to peak during the day
and dip at night, and reached annual maximums/minimums in spring-summer/fall-winter (e.g.,
Zhu et al., 2012; Lan et al., 2012), while opposite variations were observed in rural and remote
locations (e.g. Mao and Talbot 2012). Over land GOM concentrations appear to reach daily
peaks during the day and mostly below the limit of detection (LOD) at night (Mao and Talbot,
2012), whereas in marine locations nighttime GOM concentrations were found often above LOD
(Mao and Talbot, 2012). The spatiotemporal variation in PBM concentration and size distribution
appeared to be quite elusive, without generalized patterns, although more often than not large
concentrations were found in winter (e.g., Mao and Talbot, 2012).



Airborne measurements have suggested latitudinal variation in TGM with on average ~50
ppqv (~0.45 ng m$^{-3}$ in a standard atmosphere) lower in the tropics than in the polar region in
spring based on tropospheric data covering surface to 12 km altitude (Talbot et al., 2007, 2008;
Mao et al., 2010).  While TGM concentrations remained fairly constant with increasing height in
the troposphere (Banic et al., 2003; Radke et al., 2007; Ebinghaus et al., 2007; Talbot et al.,
2007, 2008; Mao et al., 2010), TGM/GEM was found to be depleted in stratospheric intrusion
(Talbot et al., 2007; Radke et al., 2007).  On the contrary, it has been postulated and modeled that
very high concentrations of GOM were in the free to upper troposphere (Holmes et al., 2006),
which has been on occasion measured with values up to 680 pg m$^{-3}$ (Lyman and Jaffe, 2011;
Brooks et al., 2014; Gratz et al., 2015; Shah et al., 2016) compared to often below 1 pg m$^{-3}$ over
land.
Such differences in spatiotemporal variations of speciated Hg were attributed to natural
and anthropogenic sources of not only Hg but also other reactive chemical compounds that are
involved in Hg cycling, meteorological conditions, and chemistry, all of which were highly
dependent on geographic locations and surrounding land surface types.  Therefore, it is highly
complex to delineate the effects of controlling factors determining observed spatiotemporal
variations of Hg concentrations.  Sprovieri et al. (2010b) reviewed the state of global mercury
measurements focusing on instrumentation and techniques, and ranges of concentration levels in
studies from different continents and oceanic regions up to 2009.  Atmospheric Hg research has
since continued to flourish, and in particular longer datasets accumulated in multiple regions
have become available for temporal variability characterization so as to understand the driving
mechanism for such variabilities.  Also of importance is the efficacy of emission reductions that
have been implemented in North America and Europe for nearly two decades and over shorter



periods in East Asia.  This paper is, different from Sprovieri et al. (2010b), aimed to provide a
global picture of spatiotemporal variations of speciated Hg using measurement-based studies in
the literature over ocean, over land, by altitude, and by latitude, and further glean insight on
important factors that could potentially contribute to the observed variations.

It should be noted that ***units were converted for a standard atmosphere*** for comparison.

One more cautionary note is that Hg data in earlier studies had coarser temporal resolution than
in more recent studies, and hence the comparisons should be viewed with this caveat in mind.
Though the earlier studies tended to have orders of magnitude larger concentrations, suggesting
at higher temporal resolution those concentrations would have been even larger.
**2. Marine Boundary Layer**

Measured TGM/GEM, GOM, and PBM concentrations in the marine boundary layer

globally were summarized in Tables S1 – S3 of the supplementary information (SI).  The MBL
studies providing these measurement data were discussed by ocean/sea.  For each ocean/sea,
spatiotemporal variations in speciated Hg and the potential causes for these variations were
summarized with respect to their ambient concentration levels, continental (including
anthropogenic) influence, hemispheric gradient, diurnal to annual cycles, and long term trends,
accompanied by discussions on potential causal mechanisms.
*2.1 TGM/GEM*

TGM and GEM in the MBL atmosphere have been measured since the late 1960s. Near

the surface in most environments, except polar springtime and Dead Sea mercury depletion
events (MDEs) when strong GEM oxidation occurs, the difference between TGM and GEM was
small to negligible (e.g., Temme et al., 2003a; Mao and Talbot, 2012).  Concentrations were
generally higher in near-coastal regions due largely to anthropogenic influence, which under



certain meteorological conditions could extend to even open oceans.  Natural emissions
including biomass burning, volcanic, and oceanic emissions were suggested to be of influence in
some studies.  It was also found that meteorological conditions could play important roles in
determining ambient concentrations of TGM/GEM via transport, PBL dynamics, and solar
radiation, especially in regions nearing emission sources such as the Mediterranean, and in
springtime Polar Regions.   Long term trends have varied over different time periods, speculated
to be associated with changing anthropogenic and natural emissions.

2.1.1 Concentration Metrics

The mean concentrations of TGM/GEM reported from various studies ranged from 1.05

ng m$^{-3}$ over the Antarctic Ocean to 2.34 ng m$^{-3}$ over the West Pacific seas, as shown in Table S1
(references therein).  The concentration averaged for each oceanic region over values reported
from the studies was the lowest 1.53 ng m$^{-3}$ over the Antarctic Ocean and the largest 2.36 ng m$^{-3}$
over the West Pacific seas (**Fig. 1a**).  The range over the Atlantic 0.05 – 29 ng m$^{-3}$ (**Fig. 1a**),
obtained from individual studies, appeared to be the largest, although the maximum
concentration was from a single event influenced by forest fires in Quebec, Canada at a long
term site in the MBL 20 km from the coast of southern New Hampshire, USA (Mao  and Talbot,
2012).  In fact, the TGM/GEM concentrations were much more variable in the MBL of the
Mediterranean Sea and its nearby seas (Table S1; references therein).

Atmospheric Hg over the *Atlantic* Ocean has been studied most extensively compared to

other oceans, largely via shipboard measurements.  Over the four decades of 1973 – 2013 from
the near-coastal to open waters over the Atlantic Ocean, concentrations of TGM/GEM ranged
from 0.05 ng m$^{-3}$ (15-minute average) in Cape Point, South Africa (Brunke et al., 2010) to 29 ng
m$^{-3}$ (5-minute average) near the shore of southern New Hampshire, USA (Mao and Talbot, 2012).





In the earliest shipboard global study of atmospheric Hg, Seiler et al. (1980) found highly
variable TGM concentrations (1 – 10 ng m$^{-3}$, 2-4 h average) averaged at 2.8 ng m$^{-3}$ between
Hamburg (54°N, 10°E) and Santo Domingo (20°N, 67°W) across the Atlantic Ocean over 11
October - 1 November 1973.  During the following 40 years, most studies reported TGM/GEM
ranging from below LOD to a few ng m$^{-3}$ and higher concentrations in near-coastal regions
(Table S1; references therein).  The *first* time Hg species measured was a one month shipboard
study over the South Atlantic Ocean during polar summer (February) 2001 by Temme et al.
(2003b).  Their measurements (5-min – 15-min average data) exhibited very small variation with
TGM averaged at 1.1(±0.2) ng m$^{-3}$ and no significant difference between TGM and GEM during
the cruise from Neumayer to Punta Arenas.  The mean concentrations over the South *Atlantic*
hovered around 1 ng m$^{-3}$ with standard deviation <0.3 ng m$^{-3}$ compared to larger mean values
(1.3 – ~3 ng m$^{-3}$) over the North Atlantic. Relatively homogeneous distributions of TGM/GEM
were observed over open waters in the South *Atlantic*.

Atmospheric Hg over the *Pacific* Ocean has been studied since the 1960s.  The oldest

data over the Pacific Ocean are from Williston (1968), in the San Francisco Bay area (Los Altos)
over a 2-year period in the early 1960s, with concentrations from the Pacific varying over 1 – 2
ng m$^{-3}$.  Over the following five decades of studies, 1 - 15-min TGM/GEM concentrations
measured over the North and South *Pacific* Ocean ranged from 0.3 ng m$^{-3}$ over 40° – 45°N in
July – September 2008 (Kang and Xie, 2011) to 7.21  ng m$^{-3}$ in the Los Angeles Port on 27 May
2010 (Weiss-Penzias et al., 2013), with generally higher concentrations near coasts and lower
ones over open oceans (Table S1; reference therein).  The distribution of TGM/GEM over the
South *Pacific* appeared to be quite heterogeneous, where Xia et al. (2010) measured TGM



averaged at 2.20±0.67 ng m$^{-3}$, a factor of 2 higher than those in Sorensen et al. (2010) that
measured a mean of 1.03 ±0.16 ng m$^{-3}$.

Over the *South China Sea, the Yellow Sea,* and other neighboring seas, located on the

Eastern Asian continental margin in the tropical-subtropical western North Pacific, adjacent to
major atmospheric Hg emission source regions, elevated concentrations of TGM/GEM were
observed with mean values varying over 2.08 – 2.62 ng m$^{-3}$ (Fu et al., 2010; Nguyen et al., 2011;
Ci et al., 2011) (Table S1). TGM concentrations measured over the *Mediterranean Sea, Adriatic*
*Sea, Dead Sea, Augusta Basin,* and *Baltic Sea* ranged from 0.4 to 11 ng m$^{-3}$ (Table S1; references
therein).

There were a few studies on Hg over the *Indian* Ocean (Soerensen et al., 2010; Xia et al.,

2010; Witt et al.; 2010; Angot et al., 2014), showing a concentration gradient of TGM with
increasing concentrations at more northern locations closer to the inter-tropical convergence
zone (ITCZ), with a mean concentration of 1.24±0.06 ng m$^{-3}$ in the Indian Ocean at latitudes
ranging from 9°S to 21°S (Witt et al., 2010).

Studies on TGM/GEM over the *Arctic* Ocean showed fairly constant concentrations of

TGM/GEM in January and August – December, MDEs in spring, and summertime annual
maximums (Lindberg et al., 2002; Aspmo et al., 2006; Sommar et al., 2010; Steffen et al., 2013;
Yu et al., 2014) . During the 1998 – 2001 Barrow Atmospheric Mercury Study (BAMS), daily
average GEM concentrations ranged from <0.2 ng m$^{-3}$ to ~3.7 ng m$^{-3}$, averaged between 1.5 – 2
ng m$^{-3}$ in January and mid-August – December (Lindberg et al., 2002). In summer 2004, 2005,
and 2012, means and ranges, well within the 1999 summertime range of Lindberg et al. (2002),
were measured by Aspmo et al. (2006), Sommar et al. (2010), and Yu et al. (2014) (Table S1).





Different concentrations of GEM over sea ice (1.81±0.43 ng m$^{-3}$) vs. ice-free (1.55±0.21
ng m$^{-3}$) Arctic Oceanic waters were measured by Sommar et al. (2010) in summer 2005.  In
spring 2009 (14 – 26 March) a mean 5-min GEM concentration of 0.59 ng m$^{-3}$ was measured
with a range of 0.01–1.51 ng m$^{-3}$ over sea ice on the Beaufort Sea near Barrow, Alaska, which
appeared to be depleted compared to annual Arctic ambient boundary layer concentrations
(Steffen et al., 2013).
In the *Antarctica*, the first study, conducted by de More et al. (1993), reported a mean
TGM concentration of 0.55 (±0.28) ng m$^{-3}$ and a range of 0.02 - 1.85 ng m$^{-3}$ (24-48 h) at Ross
Island during 1987 – 1989.  Over November 2000 – January 2001, Sprovieri et al. (2002)
reported a similar range but a mean of 0.9 (±0.3) ng m$^{-3}$, twice larger than that of de More (1993)
a decade earlier.  Similar means and ranges of TGM/GEM concentrations were measured by
Ebinghaus et al. (2002b), Temme et al. (2003b), Soerensen et al. (2010), and Xia at al. (2010).
Similar mean values but a much wider range (0.02 – 3.07 ng m$^{-3}$) were found in the multi-year
dataset in Pfaffhuber et al. (2012) (Table 1).
2.1.2 Hemispheric Difference
Hemispheric gradient over the *Atlantic* Ocean has been reported since the 1980s, with
higher concentrations in the North Atlantic attributed to anthropogenic and biomass burning
emissions (Seiler et al, 1980; Slemr et al., 1981, 1985, 1995; Slemr and Langer, 1992; Fitzgerald
et al., 1996; Lamborg et al., 1999; Temme et al., 2003a; Soerensen et al., 2010; Müller et al.,
2012).  Average TGM concentrations of 1.45 and 1.08 ng m$^{-3}$ were measured in the Northern and
Southern Hemisphere (NH, SH), respectively, in October – November 1973 (Seiler et al., 1980).
Measurements from the same cruise paths from Hamburg (54°N) and Buenos Aires (35°S) in
1977, 1978 – 1980, 1992, and 1994 consistently showed TGM hemispheric difference, 1.56±0.32



and 1.05±0.22 ng m$^{-3}$ in the NH and SH, respectively, in 1977, increased to 2.25±0.41 and
1.50±0.30 ng m$^{-3}$ followed by significant decreases to 1.79±0.41 and 1.18±0.17 ng m$^{-3}$ in 1994
(Slemr et al., 1981, 1985, 1995; Slemr and Langer, 1992).  The hemispheric difference averaged
over fall 2006 and spring 2007, documented by Soerensen et al. (2010), with a NH average of
1.32±0.16 ng m$^{-3}$ in summer and 2.61±0.36 ng m$^{-3}$ in spring, and a SH average of 1.27±0.2 ng m$^{-}$
$^{3}$, was close to the 1978 – 1980 hemispheric gradient in Slemr et al. (1985) but lower than the
1990 value in Slemr and Langer (1992).

Hemispheric gradient in TGM/GEM concentrations over the *Pacific* has been reported

with higher values in the Northern Hemisphere, mostly ascribed to its larger anthropogenic
emissions (Seiler et al., 1980; Chand et al., 2008; Xia et al., 2010; Soerensen et al., 2010).  Seiler
et al. (1980) found average TGM concentrations of 1.45 ng m$^{-3}$ and 1.08 ng m$^{-3}$ in NH and SH,
respectively, at 6-8 km altitudes over the *Pacific* Ocean in fall 1973.  A close hemispheric
gradient was found in October 1980 shipboard measurements from Fitzgerald et al. (1984) with a
constant concentration 1.5 ng m$^{-3}$ north of 4°N, a decrease to ~1 ng m$^{-3}$ south of 10°S.  Higher
concentrations but similar magnitude of hemispheric difference of TGM was measured 34 years
later in December 2007 by Xia et al. (2010) with a mean of 1.746±0.513 ng m$^{-3}$ over the North
Pacific and 1.471±0.842 ng m$^{-3}$ over the South *Indian* Ocean (Note: their cruise passed through
the South Indian instead the South Pacific).  Around the same time, Soerensen et al. (2010)
measured nearly twice lower concentrations over the South *Pacific* (1.11±0.11 ng m$^{-3}$ along the
Chilean Coast and up to 1.33±0.24 ng m$^{-3}$ near East Australia) than the North *Atlantic*
concentrations (mean values of 2.26 and 2.86 ng m$^{-3}$ over 23°N – 59°N; no measurements over
the North *Pacific* in the study) from the same study.

Studies found higher TGM concentrations up to ~2.3 ng m$^{-3}$ over *the equatorial Pacific*



in October 1980, markedly higher (>0.5 ng m$^{-3}$) than those outside this region, demonstrated to
be caused by upwelling, biological production, and anthropogenic emissions (Fitzgerald et al.,
1984; Kim and Fitzgerald, 1988).   However, Wang et al. (2014) found no sustained high GEM
concentrations indicative of persistently enhanced biotic mercury evasion from the upwelling
region over the Galápagos Islands in the equatorial Pacific during February – October 2011.
They found GEM concentrations averaged at 1.08±0.17 ng m$^{-3}$, twice lower than the earlier ones,
and significant correlation between GEM and sea surface temperature (SST).
2.1.3 Temporal Variations from Diurnal Cycle to Long-term Trend
2.1.3.1 Diurnal variation
Over the *Atlantic* Ocean, diurnal variation in TGM with daily peaks of 5 ng m$^{-3}$ at noon
and amplitude of 2-3 ng m$^{-3}$ was observed across the North and South Atlantic in Seiler et al.
(1980) whereas none in Slemr et al. (1981, 1985) and Slemr and Langer (1992).   Measurements
of TGM at Cape Point, South Africa (Brunke et al., 2010) and GEM at Appledore Island, Maine,
USA (Mao and Talbot, 2012) exhibited pronounced diurnal variation in summer with daily peaks
(minimums) before sunrise (in the late afternoon) and amplitude of 0.8 ng m$^{-3}$ and ~10 ppqv
(~0.09 ng m$^{-3}$), respectively.
Over the *Pacific*, significant diurnal variation in TGM/GEM concentrations have been
measured (Fitzgerald et al., 1984; Weiss-Penzias et al., 2003, 2013; Kang and Xie, 2011; Wang
et al., 2014) with daily peaks ranging from 0.7 ng m$^{-3}$ (5-min) over the Japan Sea (Kang and Xie,
2011) to 2.25 ng m$^{-3}$ (unknown time resolution) in the equatorial region (Fitzgerald et al., 1984).
The most pronounced diurnal variation in TGM was reported in Fitzgerald et al. (1984) with
daily amplitude of 0.7 ng m$^{-3}$ in the equatorial region (4°N – 10°S).   In contrast, Laurier et al.





(2003) found no diurnal variation during a cruise from Osaka, Japan to Honolulu, Hawaii over 1
May 2002 – 4 June 2002.

Diurnal variation in GEM over the *South China Sea* was observed in the cruise study by

Tseng et al. (2012) over May 2003 – December 2005, especially in warm seasons, exhibited
minimums before sunrise and maximums around solar noon with daily peaks reaching > 4 ng m$^{-3}$
and amplitude of ~1 ng m$^{-3}$, close to Seiler et al. (1980).  Note that this diurnal pattern is in
agreement with Fitzgerald et al. (1984) and Wang et al. (2014) but opposite of what was
observed in Weiss-Penzias et al. (2003, 2013), Brunke et al. (2010), and Mao and Talbot (2012).

Over the *Arctic* diurnal variation of GEM was observed by Lindberg et al. (2002) with

noontime minimums in spring and summer, diurnal amplitude ~2 ng m$^{-3}$ on a typical day in
January – June.  On the other hand, the shipboard measurements from Sommar et al. (2010)
suggested very small near none diurnal variation.  Similarly, no diurnal variation was found over
the *Antarctica* (Pfaffhuber et al., 2012), except one case with influence of in situ human activity.

2.1.3.2 Seasonal to Annual Variation

Annual cycles of TGM/GEM were reported over *the Atlantic* in the both hemispheres.

Annual cycles with an annual maximum in January and February (austral summer) and a
minimum in austral winter and average amplitude of 0.134 ng m$^{-3}$ were observed at Cape Point,
South Africa (Slemr et al., 2008; Brunke et al., 2010).  Opposite annual variation with higher
(lower) concentrations in winter (summer) was reported from measurements over the North
Atlantic, such as Mace Head (amplitude 0.097 ng m$^{-3}$), a remote site on the west coast of Ireland
adjacent to the North Atlantic (Ebinghaus et al., 2002a) and the Appledore Island (25 ppqv, i.e.
~0.2 ng m$^{-3}$) site in Mao and Talbot (2012).  Significant seasonal variation in NH with an annual
minimum in July and maximum in January – March and amplitude of 0.3 – 0.4 ng m$^{-3}$,was





measured in a global cruise (Soerensen et al., 2010), in close agreement with Ebinghaus et al.
(2002b; 2011), Sigler et al. (2009a), and Mao and Talbot (2012).
Average seasonal difference of 0.19 ng m$^{-3}$ GEM concentrations over the *Pacific* were
observed by Wang et al. (2014) with the highest and most variable concentrations over February
– May 2011 and the lowest and least variable in October over the Galápagos Islands during 12
November 2011 – 11 December 2011. In contrast, a *lack* of seasonal variation in GEM was
reported by Weiss-Penzias et al. (2003) using a subset of data of marine origin extracted from
one year speciated Hg data (May 2001 – May 2002) at the Cheeka Peak Observatory on the east
coast of the Pacific. This was uncharacteristic of midlatitudinal northern hemispheric sites, but
significant interannual variation was noted in this study.
Distinct annual variation in GEM over the *South China Sea* was observed in the cruise
study by Tseng et al. (2012) over May 2003 – December 2005. The winter maximum was
5.7±0.2 ng m$^{-3}$ and summer minimum 2.8±0.2 ng m$^{-3}$, 2-3 times higher than global background
levels. Difference of 0.4 ng m$^{-3}$ in seasonal average GEM was quantified with higher
concentrations in the summer than in the autumn over the Adriatic Sea (Sprovieri et al., 2010)
and a factor of two less over the Augusta Basin (Bagnato et al., 2013). The study by Obrist et al.
(2011) was the first to show the occurrence of mercury depletion events (MDEs) in midlatitudes
most frequently in summer, with GEM down to 22 ppqv (0.2 ng m$^{-3}$) in the boundary layer of the
Dead Sea, as opposed to MDEs, as commonly known, occurring in the springtime Arctic and
Antarctic only.
Annual variation of GEM over the *Indian* Ocean were reported in Angot et al. (2014)
with higher concentrations in winter (1.06±0.09 ng m$^{-3}$) and lower in summer (1.04±0.07 ng m$^{-3}$),





opposite of those at Cape Point (Slemr et al., 2008) and Galapagos Islands (Wang et al., 2014)
with annual amplitude an order of magnitude smaller.

Annual maximum concentrations of GEM occurred in summer over the *Arctic* Ocean and

frequent MDEs with GEM depleted to near zero in spring (Lindberg et al., 2002; Aspmo et al.,
2006; Cole et al., 2013; Moore et al., 2013).  Lindberg et al. (2002) observed GEM
concentrations up to 4 ng m$^{-3}$ in June 2000 compared to 1.82±0.24 ng m$^{-3}$ in summer 2004
(Aspmo et al., 2006) and 1.23±0.61 ng m$^{-3}$ in summer 2012 (Yu et al., 2014).

Seasonal variation in *Antarctic* Hg suggested large variation in TGM/GEM in spring due

to the occurrence of MDEs.  The longest continuous data record in the Antarctic started in
February 2007 at the Norwegian Antarctic Troll Research Station (TRS) in Queen Maud Land
near the Antarctic coast (Pfaffhuber et al., 2012).  Concentrations were fairly constant hovering
at ~1 (±0.07) ng m$^{-3}$ in late fall through winter and highly variable ranging from 0.02 to 3.04 ng
m$^{-3}$ averaged at 0.86 (±0.24) ng m$^{-3}$ in spring and summer (Pfaffhuber et al., 2012), close to the
values from 6 years earlier in Sprovieri et al. (2002) and Temme et al. (2003b).

2.1.3.3 Long-term Trends

North *Atlantic* long-term trends in TGM varied during different time periods of the past

decades.  An increasing rate of 1.46±0.17% yr$^{-1}$ in TGM concentrations from 1970 to 1990
(Slemr and Langer, 1992) was followed by a 22% decrease from 1990 to 1994 (Slemr et al.,
1995) according to the measurements spanning latitudes over the Atlantic from Hamburg,
Germany to Punta Arenas, Chile. In similar latitudinal coverage but over a wider longitudinal
span during three cruises in September – November 1996, December 1999 – March 2000, and
February 2001 (Temme et al., 2003a), TGM concentrations were averaged at 1.26 (±0.1) ng m$^{-3}$
ranging from 0.76 to 1.84 ng m$^{-3}$, comparable to the 1977 – 1980 (Slemr et al., 1985) and 1994



concentrations (Slemr et al., 1995) but lower than the 1990 ones (Slemr et al., 1992).  Over
September 1995 – December 2001, a slight increase (4%) in TGM was observed at Mace Head
(Ebinghaus et al., 2002a).  In the South Atlantic at Cape Point a small but significant decrease
was reported in TGM annual median from 1.29 ng m$^{-3}$ in 1996 to 1.19 ng m$^{-3}$ in 2004 (Slemr et
al., 2008).

As long-term continuous measurement data of Hg had been accumulated, studies

examined decadal trends in atmospheric TGM/GEM concentrations.  A decreasing trend of -
0.034±0.005 ng m$^{-3}$ yr$^{-1}$ in TGM was measured at Cape Point, South Africa over 1996 – 2008
(Slemr et al., 2011).  During the same time period, a statistically significant decreasing trend of -
0.028± 0.01 ng m$^{-3}$ yr$^{-1}$ (~1.6-2.0% yr$^{-1}$) in TGM over the North *Atlantic* was reported by
Ebinghaus et al. (2011) using data from Mace Head, Ireland. For the same site Weigelt et al.
(2015) presented a decreasing trend of -0.016 ± 0.002 ng m$^{-3}$ yr$^{-1}$ in monthly median marine
GEM concentrations over a longer period February 1996 to December 2013. A steep 1990–2009
decline of -0.046±0.010 ng m$^{-3}$ yr$^{-1}$ (-2.5% yr$^{-1}$) was found in TGM over the North Atlantic
(steeper than at NH land sites) but no significant decline over the South Atlantic (Soerensen et al.,
2012).  A recent comparison by Slemr et al. (2015) found smaller trends during shorter time
periods and a possible increasing trend at Cape Point for the period 2007–2013, qualitatively
consistent with the trend changes observed at Mace Head (Weigelt et al., 2015).

Over the *Arctic* Ocean, weak or insignificant declines in TGM at rates of -0.007±0.019

and 0.003±0.012 ng m$^{-3}$ yr$^{-1}$ were found at Alert and Zeppelin, respectively, during 2000 – 2009,
significantly smaller than the trends at midlatitude sites (Cole et al., 2013; Berg et al., 2013;
Ebinghaus et al., 2011; Slemr et al., 2011; Soerensen et al., 2012; Weigelt et al., 2015).
TGM/GEM concentrations over the *Antarctic* Ocean appeared to have increased from the 1980s



to the 2000s (Ebinghaus et al., 2002b; Temme et al., 2003b; Soerensen et al., 2010; Xia et al.,
2010; Pfaffhuber et al., 2012). However, no significant trend in the *Antarctic* Ocean could be
detected in mercury concentrations over 2007 – 2013 (Slemr et al., 2015).

2.1.4 Mechanisms Driving the Observed Temporal Variabilities

2.1.4.1 Causes for Episodic Higher Concentrations

It has been hypothesized that anthropogenic, biomass burning, and volcanic emissions

caused higher concentrations over open waters and near-coastal regions in many cases.  Such
influences on the atmospheric concentration of Hg was demonstrated using backward trajectories
and correlations of TGM/GEM with carbon monoxide (CO), $^{222}$Rn, black carbon, sulfur dioxide
(SO$_2$), and dimethylsulfide (DMS) (Williston, 1968; Seiler et al., 1980; Fitzgerald et al., 1981;
Fitzgerald et al., 1984; Kim and Fitzgerald, 1988; Slemr et al., 1981; Slemr et al., 1985; Slemr
and Langer, 1992; Slemr, 1996; Lamborg et al., 1998; Sheu and Mason, 2001; Laurier and
Mason, 2007; Soerensen et al., 2010; Mao and Talbot, 2012; Müller et al., 2012; Xia et al., 2010;
Chand et al., 2008; Kang and Xie, 2011; Weiss-Penzias et al., 2013; Fu et al., 2010; Nguyen et
al., 2011; Ci et al., 2011).  Some studies also suggested that oceanic evasion was an important
source contributing to higher concentrations (Seiler et al., 1980; Sigler et al., 2009b), while
others thought otherwise (Slemr et al., 1981, 1985; Slemr and Langer, 1992).  Strong
photoreduction could have caused higher TGM/GEM concentrations under favorable
meteorological conditions (Pirrone et al., 2003; Sprovieri et al., 2003; Sprovieri and Pirrone,
2008).  These influences often occurred in multitude simultaneously leading to elevated ambient
Hg concentrations.

For instance, GEM concentrations averaged at 2.86 ng m$^{-3}$ over the Sargasso Sea and the

Atlantic legs during March – April 2007 were speculated to be due to oceanic evasion and



anthropogenic influence (Soerensen et al., 2010). Mainland, ship, and volcanic emissions
appeared to elevate low concentrations of 5-min TGM in the northern Japan Sea, mostly <0.5 ng
$m^{-3}$, to ~7 ng $m^{-3}$ concurrent with peaks in CO and $SO_2$ at Nome Harbor of America (Kang and
Xie, 2011). Higher TGM concentrations over the Mediterranean Sea, Adriatic Sea, Dead Sea,
Augusta Basin, and Baltic Sea were suggested to have resulted from anthropogenic influence and
oceanic evasion (Pirrone et al., 2003). The anthropogenic contribution was corroborated in
Bagnato et al. (2013), who suggested that the basin was a receptor for Hg from intense industrial
activity with an emission flux of 0.004 t $yr^{-1}$. The role of natural emissions was underscored in
an overview of studies on Hg in the Mediterranean Sea region covering field campaigns from
2000 to 2007 (Kotnik et al., 2014). The sunny, warm and dry climate with lower amounts of
precipitation in the region was conducive to photoreduction of oxidized Hg in water column
leading to strong oceanic evasion contributing to higher TGM concentrations in the
Mediterranean Sea Basin (Pirrone et al., 2003; Sprovieri et al., 2003; Sprovieri and Pirrone,

2008).

2.1.4.2 Diurnal Variation

Nearly in all studies diurnal variation was found to be most pronounced in warm seasons,

i.e. spring and/or summer. Different combinations of oceanic emissions, photooxidation,
biological production, and meteorology were suggested to work together shaping the observed
patterns in different oceanic regions. While the pattern with daytime peaks was attributed to
oceanic emissions and biological production in sea water (Seiler et al., 1980; Fitzgerald et al.,
1984; Tseng et al.; 2012; Wang et al., 2014), the opposite pattern with daytime minimums was
associated with photooxidation and meteorological conditions (Lindberg et al., 2002; Brunke et
al., 2010; Mao and Talbot, 2012; Weiss-Penzias et al., 2003, 2013)





Over the *Atlantic* Ocean, oceanic emissions, and photooxidation were speculated to shape
the diurnal variation of TGM/GEM (Seiler et al., 1980; Brunke et al., 2010). However, Mao et al.
(2012) suggested that the predominant effect of oceanic evasion on ambient GEM concentrations
was episodic, not necessarily diurnal, because they found, among all physical parameters, the
only significant correlation GEM had was with wind speed exceeding 15 m s$^{-1}$ at a marine
location, which occurred rather sparsely. This was corroborated by Sigler et al. (2009b)
suggesting enhanced oceanic evasion at a rate of ~7 ppqv hr$^{-1}$ leading to 30 – 50 ppqv increases
in coastal and inland GEM concentrations in southern New Hampshire, USA during the April
2007 Nor'easter. Measurements of TGM at Cape Point, South Africa (Brunke et al., 2010) and
GEM at Appledore Island, Maine, USA (Mao and Talbot, 2012) exhibited pronounced
summertime diurnal variation with daily peaks (minimums) before sunrise (in the late
afternoon),which was speculated to be caused by daytime GEM oxidation by halogen radicals in
the marine environment.
Over the *Pacific*, significant diurnal variation in TGM/GEM concentrations have been
linked to biological production, photochemistry, and meteorology (Fitzgerald et al., 1984; Weiss-
Penzias et al., 2003, 2013; Wang et al., 2014). The most pronounced diurnal variation in TGM
in the equatorial area (4°N – 10°S) was demonstrated to be caused by biological production
(Fitzgerald et al., 1984). Diurnal variation with significantly higher nighttime concentrations
near the coast of Los Angeles was ascribed to the nighttime urban outflow (Weiss-Penzias et al.,
2013). Strong daytime photooxidation was speculated to have contributed to the marked diurnal
variation with nighttime maximums in summer and spring in Weiss-Penzias et al. (2003) and
Wang et al. (2014), respectively. In the study by Laurier et al. (2003) the lack of diurnal
variation was speculated to be caused by continuous evasion from surface water.



GEM diurnal variation with minimums before sunrise and maximums around solar noon

over the *South China Sea*, especially in warm seasons, was linked to oceanic evasion, which was
supported by the concurrent measurements of dissolved elemental Hg (Tseng et al., 2012).  The
100 m MBL height assumed for estimation appeared to be too low, indicating that other factors
may have contributed to the diurnal pattern.

Noontime GEM minimums in spring and summer over the Summertime *Arctic* suggested

photooxidation of GEM (Lindberg et al., 2002).  On the other hand, the very small near none
diurnal variation in GEM manifested in the shipboard measurements of Sommar et al. (2010)
was speculated to result from low in situ oxidation of GEM.  No diurnal variation was found
over the *Antarctica* due possibly to lack of diurnally varying sources and sinks (Pfaffhuber et al.,
2012), except one case with in situ human activity.

2.1.4.3 Seasonal to Annual Variation

Annual cycles of TGM/GEM in the MBL differed in various oceanic regions and were

suggested to be driven predominantly by oceanic evasion, biomass burning, anthropogenic
emissions, interhemispheric flux, and meteorological conditions (Slemr et al., 2008; Ebinghaus
et al., 2002a,b; Sigler et al., 2009a; Brunke et al., 2010; Soerensen et al., 2010; Mao and Talbot,
2012; Angot et al., 2014; Wang et al., 2014).  Over the Atlantic annual cycles of TGM/GEM
with an annual maximum in summer and a minimum in winter at Cape Point, South Africa was
hypothesized to be driven predominantly by the total emissions from oceans, biomass burning,
and anthropogenic activities (Brunke et al., 2010), and the interhemispheric flux (Slemr et al.,
2008; Brunke et al., 2010).  *Opposite annual variation* with higher (lower) concentrations in
winter (summer) was proposed to be largely determined by meteorology (Ebinghaus et al., 2002a,
2011) and photochemical oxidation of GEM (Mao and Talbot, 2012).  The same annual cycle





with higher concentrations in winter over the *Indian* Ocean (Angot et al., 2014), opposite of
those at Cape Point (Slemr et al., 2008) and Galapagos Islands (Wang et al., 2014), was
speculated to be a result of long range transport of air masses originated from biomass burning
emissions in southern Africa during the winter months (July – September), and low GEM
associated with southerly polar and marine air masses from the remote southern Indian Ocean.
Higher concentrations of GEM in the summer over the *Adriatic Sea* (Sprovieri et al., 2010) and
over the *Augusta Basin* (Bagnato et al., 2013) were suggested to be caused by the stagnant
meteorological conditions in the former study and enhanced evasion from sea water in the latter.

Midlatitudinal MDEs were first reported by Obrist et al. (2011), which occurred in the

MBL of the Dead Sea.  The MDEs in the Dead Sea boundary layer were observed to be often
concurrent with varying concentrations of bromine oxide (BrO) and high temperatures up to
45°C. Such high temperatures seemed to be contradictory to the general understanding that Br-
initiated GEM oxidation tends to go forward under very cold conditions at temperature < -40°C.
The authors suggested that Br species were the major oxidants of GEM during depletion events,
in spite of constant high temperatures accompanied by sometimes low BrO concentrations.

Springtime large variation in *Arctic* and *Antarctic* TGM/GEM was suggested to be a

result of the occurrence of MDEs.  Polar MDEs have been generally linked to reactive Br-
initiated GEM oxidation in spring when Br explosion occurs producing abundant reactive Br
(Schroeder et al., 1998; Ebinghaus et al., 2002b; Lindberg et al., 2002; Temme et al., 2003b;
Mao et al., 2010; Steffen et al., 2013; Moore et al., 2014).  For Antarctic MDEs, Ebinghaus et al.
(2002b) found a strong positive correlation between TGM and $O_3$ over August – October,
accompanied by enhanced Global Ozone Monitoring Experiment (GOME) column BrO.
Compared to Arctic MDEs, the first Antarctic MDE occurred about 1-2 months earlier, probably





due to the lower latitude of the monitoring site and sea ice, the former allowing earlier sunrise
and the latter conducive to Br/BrO formation.  Temme et al. (2003b)  found that the air masses
reaching the station during MDEs had a maximum contact with sea ice (coverage >40%) over
the South Atlantic Ocean, which was speculated to contain abundant reactive Br, released from
sea salt associated with sea ice or sea salt aerosols.

Summertime annual maximums of GEM over the *Arctic* and *Antarctic* Ocean were

generally associated with maximum exposed sea water after snow/ice melt (Lindberg et al., 2002;
Aspmo et al., 2006; Soerensen et al., 2010; Cole et al., 2013; Moore et al., 2014) and were also
in the Arctic with riverine input (Fischer et al., 2012) as well as with enhanced reduction by high
chromophoric dissolved organic matter (CDOM) in river runoff (Yu et al., 2014). Soerensen et al.
(2010) found a temperature decrease and wind coming along the Antarctica coast partly covered
with sea ice corresponding to increases in GEM concentrations, which were speculated to be
from reemission from snow covered surface or the release of dissolved gaseous mercury (DGM)
in supersaturated environments exposed after ice melt.  Lindberg et al. (2002) associated
observed GEM concentrations up to 4 ng m$^{-3}$ in June with enhanced evasion of GEM dissolved
and from GOM reduction in snow.  Aspmo et al. (2006) linked the summertime annual peak of
GEM to >70% sea ice, possibly related to biotic reduction leading to higher concentrations of
DGM in sea water binding more Hg and hence larger evasion  in open leads in the sea ice.  This
hypothesis was further supported by Moore et al. (2014), who found coastal AMDEs in the
springtime Arctic linked to sea-ice dynamics using backward trajectories, as well as by the
model simulations of Dastoor and Durnford (2014).  A different mechanism of riverine
contribution was hypothesized in Fischer et al. (2012) using an atmosphere-ocean coupled model.
Yu et al. (2014) observed high TGM concentrations concurrent with low salinity, CO, and high



CDOM over the ice-covered central Arctic Ocean and speculated that the relatively high CDOM
concentrations associated with river runoff could enhance $Hg^{2+}$ reduction.  Moreover they related
the summer monthly variability in TGM concentrations to less chemical loss.

2.1.4.4 Long-term Trends

Varying trends in TGM/GEM different periods of the past decades were speculated to be

due largely to changes in anthropogenic emissions and at times natural emissions.  A case in
point is the 1970 – 1990 1.46±0.17% $yr^{-1}$ increasing rate of TGM concentrations (Slemr and
Langer, 1992) followed by a 1990 – 1994 22% decrease (Slemr et al., 1995) shown in the
measurements over the Atlantic from Hamburg, Germany to Punta Arenas, Chile. These trends
were attributed to changing anthropogenic emissions and possibly decreased natural emissions
associated with climate cooling in the wake of Pinatubo eruption.  Ebinghaus et al. (2002a)
suggested that a slight increase (4%) in TGM at Mace Head over September 1995 – December
2001likely resulted from *increased* anthropogenic emissions.  Soerensen et al. (2012) found that
sea surface water elemental Hg concentrations were decreasing at a rate of -5.7% $yr^{-1}$ since 1999,
which might explain the steep 1990–2009 decline of -0.046±0.010 ng m⁻3 $yr^{-1}$ (-2.5% $yr^{-1}$) in
TGM over the North Atlantic.  A recent comparison by Slemr et al. (2015) suggested that the
opposing trends over the periods of 1996–2004 (increasing) and 2004–2007 (possibly decreasing)
might have led to smaller trends at shorter time periods and an increasing trend at Cape Point for
the period 2007–2013.

Three *hypotheses* were made to explain these decadal decreasing trends.  First, the global

decreasing trend was caused by decreased reemission of legacy mercury as a result of a
substantial shift in the biogeochemical cycle of Hg through the atmospheric, oceans, and soil
reservoirs, although exactly what may have caused this shift remained unexamined (Slemr et al.,



2011). Conflicting evidence was found by Ebinghaus et al. (2011) for worldwide changing
anthropogenic emissions, and hence the decreasing trends could not simply be attributed to
decreasing anthropogenic emissions in some regions. They hypothesized that the decreasing
trend was linked to increasing tropospheric $O_3$, and yet this speculation was negated by the
plausibility of GEM oxidation by $O_3$ in the atmosphere. The third hypothesis, developed by
Soerensen et al. (2012), was that, based on atmosphere-ocean coupled model simulations, the
decreasing trend in TGM over the North Atlantic was caused by decreasing North Atlantic
oceanic evasion driven by declining subsurface water Hg concentrations resulting from reduced
Hg inputs from rivers and wastewater and from changes in the oxidant chemistry of the
atmospheric MBL.
***2.2 GOM and PBM***
2.2.1 Concentration Metrics

The mean concentrations of GOM from individual studies varied from below LOD in

several studies to 4018 pg m$^{-3}$ (1-h) in the Dead Sea MBL from Obrist et al. (2011) and Moore et
al. (2013), as shown in Table S2 (references therein). The GOM concentration averaged for each
oceanic region based on values from the literature varied from 3 pg m$^{-3}$ over the Atlantic to 40 pg
m$^{-3}$ over the Antarctic Ocean (**Fig. 1b**), and the largest range 0.1 – 4018 pg m$^{-3}$ was over the
Mediterranean Sea and its neighboring seas (**Fig. 1b**). Note that the small ranges in other
oceanic MBL did not necessarily indicate less variability in GOM but merely a result of limited
measurement data available (Table S2; references therein).

The mean concentrations of PBM from individual studies varied from below LOD in

several regions to 394 pg m$^{-3}$ (1-h) over the Beaufort Sea (Steffen et al., 2013) (Table S3;
references therein). The PBM concentration averaged for each oceanic region based on values in





the literature varied from 0.6 pg m$^{-3}$ over the Indian to 394 pg m$^{-3}$ over the Arctic Ocean (**Fig.**
**1c**).  Due to limited numbers of studies in the Arctic, Antarctic and Indian Ocean MBL, no
ranges were provided for each one of them. The ranges for the six oceans were not comparable
as very few studies were available in some of them.  However, the few studies available
indicated that PBM concentrations were in most cases smaller and less variable than GOM.

The earliest shipboard measurements of GOM showed dimethyl mercury (DMM)

concentrations of ⁓0.1 ng m$^{-3}$ comprising ⁓10% of TGM in clean marine air as opposed to 0.4 –
15.3 ng m$^{-3}$ in polluted air during the 1977 cruise (Slemr et al., 1981), and ranging between 0.02
and 0.12 ng m$^{-3}$ (6-h) comprising <2% of TGM, during the 1978 – 1981 cruises across the
*Atlantic* between Hamburg (50°N) and Buenos Aires (35°S) (Slemr et al., 1985). From the late
1990s to the 2010s generally GOM concentrations, instead of DMM, were measured and were
mostly orders of magnitude smaller, except during MDEs when GOM concentrations could be
on the order of magnitude of $10^2$ pg m$^{-3}$ (Table S2; references therein).

Same as GEM, GOM concentrations tended to be higher over the North than the South

*Atlantic* and in near-coastal regions than open waters, and continental influence was detected at
times over open waters (Temme et al., 2003b; Mason et al., 2001; Sheu and Mason, 2001; Mason
and Sheu, 2002; Aspmo et al., 2006; Laurier and Mason, 2007; Sigler et al. 2009b; Mao and
Talbot, 2012).  1-h GOM concentrations of 1 – 30 pg m$^{-3}$ over the *South* Atlantic Ocean from
Neumayer to Punta Arenas in February 2001 (Temme et al., 2003b) were 1 – 2 orders of
magnitude smaller than the concentrations (1.38±1.30 pmol m$^{-3}$, i.e. ~300±280 pg m$^{-3}$) near
Bermuda in September and December 1999 and March 2000 (Mason et al., 2001).  However, at
around the same time average values almost an order of magnitude smaller were reported at
Bermuda (50±43 pg m$^{-3}$, a few pg m$^{-3}$ to 128 pg m$^{-3}$) (Mason and Sheu, 2002) and at a US mid-





Atlantic coastal site (40 pg m$^{-3}$) (Sheu and Mason, 2001).  In comparison, at higher northern
latitudes (54°N – 85°N), GOM concentrations averaged at 2.5 pg m$^{-3}$ varying from below LOD
to 22 pg m$^{-3}$ were comparable  to those over the South Atlantic.  In the late 2000s at a North
Atlantic MBL site 25 km off the southern New Hampshire, US, GOM was averaged at 0.4 ppqv
(~3.6 pg m$^{-3}$) (0 – 22 ppqv, i.e. 0 – 196 pg m$^{-3}$, 2-h) for May – August 2007 (Sigler et al. 2009b)
and very close values from the 2007 –2010 dataset at the same site (Mao and Talbot, 2012).
These values were close to the open water and higher latitude concentrations (Aspmo et al., 2006;
Laurier and Mason, 2007), but one to two orders of magnitude lower than the early 2000s
measurements at close latitudes (Mason et al., 2001; Sheu and Mason, 2001; Mason and Sheu,

2002).

PBM concentrations (Table S3; references therein) of similar magnitude was measured

with an average of 1.9±0.2 pg m$^{-3}$ over the May-June 1996 South and equatorial *Atlantic* cruise
(Lamborg et al., 1999) and 1.3 ± 1.7 pg m$^{-3}$ (<0.5 pg m$^{-3}$ (LOD) to 5.2 pg m$^{-3}$) in Bermuda, 30-40
times smaller than the concurrent weekly averaged GOM concentrations (Mason and Sheu, 2002;
Sheu, 2001).  At higher North Atlantic latitudes, PBM concentrations were averaged at 2.4 pg m$^{-}$
$^{3}$, very close to the concurrent average GOM concentrations but with a factor of 4 smaller
varying range (below MDL to 6.3 pg m$^{-3}$) in summer 2004 (Aspmo et al., 2006).  Mao and
Talbot (2012) reported PBM concentrations varying from 0.09 ppqv (0.8 pg m$^{-3}$) in winter 2010
to 0.52 ppqv (4.6 pg m$^{-3}$) in summer 2010 for the time period of spring 2009 to summer 2010.

During the 2000s decade, concentrations of GOM over the *Pacific* decreased by around a

factor of 2 from the mean value of 9.5 pg m$^{-3}$ over open waters in 2002 (Laurier et al., 2003) to
around 4 pg m$^{-3}$ at a remote Japanese site downwind of major Asian source regions in spring
2004 (Chand et al., 2008) and in the equatorial region in 2011 (Wang et al., 2014) (Table 2;



references therein). The maximum concentration from a decade of studies was 700 pg m$^{-3}$ (3-h), ,
measured in air masses originated from upper air over the Pacific (Timonen et al., 2013), about
two orders of magnitude larger than what Chand et al. (2008) and Laurier et al. (2003) reported.

PBM concentrations over the *Pacific* reached up to 17 pg m$^{-3}$ and mean values were three

times larger downwind of East Asia (3.0±2.5 pg m$^{-3}$) than in the equatorial Pacific MBL (Chand
et al., 2008; Wang et al., 2014) (Table S3). Chand et al. (2008) found PBM concentrations
comparable to GOM.

In the southern *Indian* Ocean, very low GOM and PBM concentrations, averaged at 0.34

(<LOD (0.28 – 0.42 pg m$^{-3}$) – 4.07 pg m$^{-3}$) and 0.67 pg m$^{-3}$ (<LOD – 12.67 pg m$^{-3}$), respectively,
were measured by Angot et al. (2014) over two years from a remote location, Amsterdam Island.
These concentrations were at the lower end of the range of MBL measurements from over the
Atlantic and the Pacific (Laurier et al., 2003; Temme et al., 2003b; Laurier and Mason, 2007).

Measurements over the *Mediterranean Sea and its neighboring seas* generally showed

much higher concentration levels than over the Atlantic, Pacific, and Indian Ocean, with GOM
ranging from 0.1 pg m$^{-3}$ over the Adriatic (Sprovieri and Pirrone, 2008) to 4018 pg m$^{-3}$ over the
Dead Sea (Obrist et al., 2011) (Tables S2 &S3; references therein). Frequency distributions of
24-hour average GOM and PBM concentrations from Palma de Mallorca, a site situated in the
Mediterranean MBL, exhibited log-normal distributions with the maximum frequency at around
59 and 48 pg m$^{-3}$, respectively (Pirrone et al., 2003). One of the major findings from Sprovieri et
al. (2003) was constant presence of GOM averaged at 7.9±0.8 pg m$^{-3}$ in the MBL over a 6000
km long cruise path around the Mediterranean Sea. In a one year dataset of 2008, Beldowska et
al. (2012) showed 24-h PBM concentrations varied over 2 – 142 pg m$^{-3}$ averaged at 20 (±18) pg
m$^{-3}$ with 93% on average in the coarse fraction (>2 μm) over the southern *Baltic* Sea.



In springtime *Arctic*, the highest concentrations of GOM at 900 – 950 pg m$^{-3}$ were

observed during the 1998 – 2001 Barrow Atmospheric Mercury Study (BAMS).  Very high

springtime PBM concentrations (mean 394 pg m$^{-3}$, 47 – 900 pg m$^{-3}$, 1-h) were reported over

Beaufort Sea sea ice by Steffen et al. (2013).  This was an order of magnitude higher than

concurrent GOM concentrations (mean 30 pg m$^{-3}$, 3.5 – 104.5 pg m$^{-3}$) and even larger than those

in temperate regions, where particle concentrations tended to be. In comparison, Sommar et al.

(2010) found very low GOM and PBM over the summertime Arctic Ocean.

Two *Antarctic* DMM measurement studies conducted by de More et al. (1993) and

Pongratz and Heumann (1999) differed by two orders of magnitude with a mean of 0.04 (±0.06)

641        ng m$^{-3}$ over a range of 0 – 0.63 ng m$^{-3}$ (24-48 h) at Ross Island from the former, speculated to be

under anthropogenic influence and a mean of 6 pg m$^{-3}$ over a range of <4 – 9 pg m$^{-3}$ over the

Antarctic Ocean from the latter (Table 2).  Total 2-h GOM concentrations ranged over 10.5 –

334 pg m$^{-3}$ averaged at 116.2(±77.8) pg m$^{-3}$ in 2000 spring – summertime in Terra Nova Bay

(Sprovieri et al., 2002), and a similar range was also observed by Temme et al. (2003b) at the

Neumayer Station in summer 2001 (Table 2).  A range of 30 – 140 pg m$^{-3}$ (80-min) for peaks of

GOM over the Antarctic Ocean in summer 2007 (Soerensen et al., 2010) coincided with small

peaks of GEM, unlike Sprovieri et al. (2002) and Temme et al. (2003b) who also saw high peaks

of GOM from the Antarctic Ocean but were anti-correlated with GEM.  Concentrations of 1-h

PBM over the *Antarctic* Ocean from Temme et al. (2003b) varied over 15 – 120 pg m$^{-3}$, a range a

factor of 3 smaller than that of concurrent GOM and tracking GOM well only at a lower level.

2.2.2 Hemispheric Difference

Hemispheric gradient has been measured in both GOM and PBM since the early 1980s

(Slemr et al., 1985; Soerensen et al., 2010).  In the first shipboard study, Slemr et al. (1985)



655 derived PBM concentrations of 0.013 ± 0.018 and 0.007 ± 0.004 ng m$^{-3}$ over the North and

656 South *Atlantic* Ocean, respectively, from the Hg concentrations in rain water.  About three

657 decades later Soerensen et al. (2010) reported hemispheric difference of GOM with a NH

658 average of 0.3±3 pg m$^{-3}$ in summer and 0.8±2 pg m$^{-3}$ in spring, and a seasonally invariable SH

659 average of 4.3±0.14 pg m$^{-3}$.

660 2.2.3 Temporal Variations from Diurnal to Long-term Trend

661 2.2.3.1 Diurnal Variation

662 While some studies found a lack of diurnal variation in GOM (Sheu and Mason, 2001;

663 Aspmo et al., 2006; Temme et al., 2003b), many studies reported pronounced diurnal variation in

664 various oceanic regions (Mason et al., 2001; Mason and Sheu, 2002; Lindberg et al., 2002;

665 Laurie et al., 2003; Sprovieri et al., 2003, 2010; Laurier and Mason, 2007; Mao et al., 2008;

666 Chand et al., 2008; Sigler et al., 2009b; Soerensen et al., 2010; Mao and Talbot, 2012; Wang et

667 al., 2014).  In only one out of seven 24-hr GOM sampling sessions did Sheu and Mason (2001)

668 find diurnal variation of GOM, with daily peaks at noon, below LOD at night and amplitude of

669 nearly 100 pg m$^{-3}$.  The studies reporting distinct diurnal variation over the Atlantic showed

670 consistent daytime peaks and nighttime minimums, with amplitude values varying from 0.27 pg

671 m$^{-3}$ in winter 2010 near the coast of southern New Hampshire, USA (Mao and Talbot, 2012)

672 to >80 pg m$^{-3}$ from Barbados via Bermuda to Baltimore, Maryland, USA (Mason and Sheu, 2002;

673 Laurier and Mason, 2007).  Distinct diurnal variation in GOM was also measured over the

674 *Pacific* Ocean with noon - afternoon maximums and nighttime minimums and amplitude > 80 pg

675 m$^{-3}$ (Laurier et al., 2003; Chand et al., 2008; Wang et al., 2014).  Over the *Mediterranean* Sea

676 and its neighboring seas diurnal variation of GOM concentrations was shown in most studies

677 with daily peaks at noon and amplitude up to 35 pg m$^{-3}$ (Sprovieri et al., 2003; Sprovieri et al.,





2010). For the springtime Arctic Lindberg et al. (2002) measured noontime maximums of GOM
up to 900 – 950 pg m$^{-3}$ and near zero concentrations at night.

The diurnal pattern of PBM concentrations, measured using a Tekran speciation unit, at a

midlatitude North *Atlantic* MBL site close to southern New Hampshire, USA was in general not
consistent between seasons and years with seasonally averaged daily peaks 0.2 – 0.7 ppqv (1.7 –
6.2 pg m$^{-3}$) at varying time of a day (Mao and Talbot, 2012). The Tekran PBM instrument
measures PBM on particles < 2.5 µm, which is naturally a fraction of total atmospheric PBM.
Using a 10-stage impactor, Feddersen et al. (2012), perhaps the first to study the size distribution
of PBM in MBL, reported PBM concentrations (up to 0.25 ppqv, i.e. 2.2 pg m$^{-3}$, in 3.3 – 4.7 µm)
in ten size fractions (<0.4 µm to >10 µm) for the same MBL location from Mao and Talbot
(2012), and found a diurnal cycle with daily maximums at around 16:00 UTC (noon local time)
and minimums around sunrise.

2.2.3.2 Seasonal to Annual Variation

Several studies reported distinct seasonal variation in GOM with higher concentrations in

warmer months and lower in colder months (Mason et al., 2001; Mason and Sheu, 2002; Pirrone
et al., 2003; Laurier and Mason, 2007; Sigler et al., 2009a; Sprovieri et al., 2010; Soerensen et al.,
2010; Mao and Talbot, 2012; Angot et al., 2014). For instance, Mason et al. (2001) found GOM
concentrations elevated in September (2.54 – 6.86 pmol m$^{-3}$) compared to those in December and
March (0.23 – 2.68 pmol m$^{-3}$) near Bermuda. At the midlatitude North *Atlantic* MBL site near
southern New Hampshire, USA, a fairly flat baseline with negligible annual variation in GOM
was observed in a three year dataset, with more variability in higher mixing ratios and seasonal
median values ranging from 0.03 ppqv (~0.27 pg m$^{-3}$) in winter 2010 to 0.55 ppqv (~4.9 pg m$^{-3}$)
in summer 2007 (Mao and Talbot, 2012). The PBM measurements using a 10-stage impactor

701 from Feddersen et al. (2012) showed distinct seasonal variation with 50-60% of PBM in coarse

702 fractions, 1.1 – 5.8 μm, composing largely of sea salt aerosols at both sites in summer and 65%

703 in fine fractions at the coastal site in winter.

704   In the equatorial *Pacific*, seasonal variation in PBM concentrations was observed with an

705 average of 1.1±1.1 pg m$^{-3}$ in October and below LOD (0.42 pg m$^{-3}$) in June (Wang et al., 2014).

706 In the southern *Indian* Ocean, a slight but significant seasonal variation was found in GOM

707 concentrations averaged at 1.34±0.45 pg m$^{-3}$ in winter vs. 1.58±0.35 pg m$^{-3}$ in summer, while a

708 seasonal trend in PBM with significantly higher concentrations in winter than in summer

709 (2.18±1.56 ng m$^{-3}$ vs. 1.79±1.15 pg m$^{-3}$) (Angot et al., 2014).

710   Over the *Mediterranean* Sea and its neighboring seas, seasonal variation in GOM

711 concentrations was found, 31.5±39.2 pg m$^{-3}$ in November, 40.4±43 pg m$^{-3}$ in February,

712 52.3±43.9 pg m$^{-3}$ in May, and 32.3±17.8 pg m$^{-3}$ in July (Pirrone et al., 2003), and the fall 2004

713 and summer 2005 campaigns experienced no production of GOM and little variation in GOM in

714 the fall and very high concentrations varying over 21 – 40 pg m$^{-3}$ in the summer (Sprovieri et al.,

715 2010a).  In the *Dead* Sea MBL, AMDEs resulting in 1-h GOM up to 700 pg m$^{-3}$occurred more

716 frequently in the summer (20 of 29 days) than in winter (8 of 20 days), the majority of which

717 were not concurrent with ozone depletion events (ODEs) (Obrist et al., 2011; Moore et al., 2013).

718 Two studies observed seasonal variation in PBM.  Sprovieri et al. (2010a) found PBM

719 concentrations on average more than a factor of 2 higher during high Hg episodes in the fall than

720 during the summertime ones over the Mediterranean Sea.  Beldowska et al. (2012) measured an

721 average 24-h PBM of 15 pg m$^{-3}$ and a 3 – 67 pg m$^{-3}$ range in the non-heating season compared to

722 an average of 24 pg m$^{-3}$ and a range of 2 – 142 pg m$^{-3}$ in the heating season.



In the Arctic MBL, several hundreds of pg m$^{-3}$ GOM concentrations were observed in
spring (Lindberg et al., 2002; Steffen et al., 2013), while in summer very low GOM and PBM
concentrations were measured (Sommar et al., 2010). Different from the Arctic, summertime
GOM concentrations over the *Antarctic* seemed to be orders of magnitude larger (Sprovieri et al,
2002; Temme et al., 2003b; Soerensen et al., 2010).
2.2.4 Mechanisms Driving the Observed Temporal Variabilities
2.2.4.1 Factors Causing Episodic High and Low Concentrations
Long range transport of air masses with high PBM concentrations of terrestrial origin was
suggested due to elevated crustal enrichment factors in the PBM samples (Lamborg et al., 1999).
An episode of high GOM concentrations coincided with a passing hurricane, which led to
speculation that downward mixing of air aloft with higher GOM (Prestbo, 1997) might have
contributed to those high concentrations (Mason and Sheu, 2002). Mason and Sheu (2002)
found low GOM concentrations concurrent with high humidity (e.g., fog) and rainfall but highest
concentrations on the day after such events if temperatures were elevated. High nighttime
concentrations of GOM in the Mediterranean Basin were observed in anthropogenic plumes
identified using backward trajectories (Sprovieri et al., 2010). The GOM concentrations in air
masses of marine origin at a site on the East Pacific coast were unusually high ranging over 200
– 700 pg m$^{-3}$ (Timonen et al., 2013). The high GOM concentrations were thought to be
partitioned back from the PBM that was accumulated on aqueous super-micron sea salt aerosols
in the MBL when being lofted above the MBL, and an anticorrelation between GOM and GEM
was found in air masses of marine origin indicating strong in-situ oxidation of GEM.
2.2.4.2 Diurnal Variation
The lack of diurnal variation observed at a US eastern seaboard coastal location was



speculated to result from diverse air masses with different concentrations converging at the
location leading to the removal of diurnal variation in GOM (Sheu and Mason, 2001).  In another
case at higher latitudes it was thought be due to low solar radiation ($<200$ W m$^{-2}$) (Aspmo et al.,
2006).  The majority of the studies reporting significant diurnal variation in GOM attributed the
diurnal pattern with daytime peaks and nighttime minimums to photooxidation, loss via dry
deposition, and oceanic evasion, which was backed up by modeling studies (Hedgecock et al.,
2003, 2005; Laurier et al., 2003; Selin et al., 2007; Strode et al., 2007).
It was generally found that GOM concentrations were positively correlated with solar
radiation flux and anticorrelated with relative humidity and at times with $O_3$ (Mason and Sheu,
2002; Laurier and Mason, 2007; Soerensen et al., 2010; Mao et al., 2012). Mason and Sheu
(2002) and Laurier and Mason (2007) pointed out that the correlation between GOM and UV
radiation flux indicated photochemical processes, and the anticorrelation between GOM and $O_3$
was caused by processes destroying $O_3$ and producing GOM, with an emphasis on oxidation
reactions in the presence of deliquescent sea salt aerosols based on the laboratory experimental
study by Sheu and Mason (2004).  The fact that the daytime peaks in GOM over the *Pacific*
increased with less wind speed, which was conducive to less dry depositional loss, and strong
UV radiation suggested that GOM was produced in situ via photochemically driven oxidation
(Laurier et al., 2003; Chand et al., 2008).  Chand et al. (2008) estimated the magnitude of GOM
close to the amount produced from the reaction of GEM + OH alone.  Mao and Talbot (2012)
speculated unknown production mechanism(s) of GOM in the nighttime MBL keeping the levels
above the LOD.  Positive correlation was found between GOM/PBM and temperature, indicating
possible temperature dependence of certain oxidation reactions and gas-particle partitioning
(Mao et al., 2012). Mao et al. (2012) also found anti-correlation between GOM/PBM and wind



speed, which was not found at the coastal and inland locations, indicating enhanced loss via
deposition caused by faster wind speed over water.

The diurnal pattern of PBM, measured using a Tekran speciation unit, in general was not

consistent from season to season as found in Mao and Talbot (2012), which indicated more
complicated processes than photochemistry involved in PBM budgets.  However, for the same
MBL location in Mao and Talbot (2012), Feddersen et al. (2012) found diurnal variation in 10-
stage impactor PBM measurement data and speculated that GEM oxidation drove the PBM
daytime maximum at around 16:00 UTC (noon local time) and deposition to aerosol surface
without replenishment led to the minimum around sunrise.  In the same study, the large peaks of
PBM appeared to be of continental origin.

2.2.4.3 Seasonal to Annual Variation

Larger concentrations of GOM in spring and/or summer were generally associated with

stronger photooxidation, biological activity, biomass burning, oceanic, and anthropogenic
emissions, whereas low concentrations could be due to wet deposition in the MBL of various
oceanic regions (Lindberg et al., 2002; Mason and Sheu, 2002; Temme et al., 2003b; Pirrone et
al., 2003; Sprovieri et al., 2003; Hedgecock et al., 2004; Laurier and Mason, 2007; Sprovieri and
Pirrone, 2008; Sprovieri et al., 2010; Soerensen et al., 2010; Obrist et al., 2011; Mao et al., 2012;
Angot et al., 2014; Wang et al., 2014).  The positive correlation between GOM concentrations
and solar radiation was used to explain warm season maximums of GOM based on the same line
of reasoning that was used to explain daytime peaks of GOM (Mason and Sheu, 2002; Pirrone et
al., 2003; Mao et al., 2012).

To explain the annual maximum GOM concentration in October over the *equatorial*

*Pacific*, Wang et al. (2014) included iodine in a two-step mercury oxidation mechanism, where


BrHgI was hypothetically formed, helped to reconcile the modelled GOM with the observations.
The authors mentioned that $HO_2$ and/or $NO_2$ aggregation with HgBr from Dibble et al. (2012)
would be another possibility and further suggested that a major process in representing Hg
oxidation is missing in current models.

In the southern *Indian* Ocean, Angot et al. (2014) speculated that very low levels of

GOM and PBM were likely due to very frequent scavenging drizzle, whereas high GOM events
in summer were associated with enhanced photochemistry and biological activity and high PBM
events in winter were related to peaking southern African biomass burning.

Over the *Mediterranean* Sea and its neighboring seas, it was generally thought that

meteorological conditions combined with anthropogenic, oceanic, and biomass emissions could
affect GOM and PBM concentrations and subsequently their seasonal variation (e. g. Pirrone et
al., 2003; Sprovieri et al., 2003; Hedgecock et al., 2004; Sprovieri and Pirrone, 2008).  For
instance, the seasonal contrast of no production and little variation in GOM in fall 2004 and very
high concentrations in summer 2005 was due likely to weather conditions (e.g., large liquid
water content, rainy, overcast) in fall 2004 and strong oxidation in summer 2005 under dry,
sunny conditions associated with a prevailing stagnant high pressure system over the region
(Sprovieri et al., 2010).  Sensitivity box model simulations suggested that the Hg + Br controlled
the production rate of GOM without contributions from the oxidation reactions by $O_3$ and OH
and that HgBr was quickly converted to GOM.  In the same study it was brought to attention that
biomass burning and ship emissions in the region were not included in the emission inventory
but could be important to ambient concentrations (Sprovieri et al., 2010).

In the Dead Sea MBL, more frequent occurrences of MDEs in the summer (20 of 29 days)

than in winter (8 of 20 days) were linked to higher BrO concentrations in summer (Obrist et al.,



2011).  It was speculated that the strong MDEs, despite high temperature and sometimes low
BrO concentrations, were caused by Br-initiated oxidation of GEM based on their box model
results.  There is apparent discrepancy between our theoretical understanding of the conditions
required for Br-initiated GEM oxidation and the real atmospheric conditions in the summertime
Dead Sea MBL.

Two studies observed seasonal variation in PBM.  Sprovieri et al. (2010a) found that

PBM concentrations on average were more than a factor of 2 higher during high Hg episodes in
the fall than during the summertime ones over the Mediterranean Sea due to anthropogenic
influence.  Beldowska et al. (2012) suggested that the higher concentrations in winter were a
result of mild temperatures and high relative humidity in winter being conducive to Hg
adsorption on the surface of coarse particles as well as condensation and coagulation of fine
particles, while during the warm season the strong influence of industrial sources led to higher
PBM concentrations on working days than on weekends.

Lindberg et al. (2002) found that springtime *Arctic* maximum concentrations of GOM at

900 – 950 pg m$^{-3}$ corresponded to open leads over sea ice and an extensive area of elevated BrO
concentrations under the calmest conditions and strongest UV radiation.  Over Beaufort Sea sea
ice in spring 2009 lower GOM compared to PBM concentrations and larger PBM concentrations
than those in temperate regions were speculated to be caused by low temperatures and the
availability of sea salt and sulfate aerosols, as well as ice crystals around the sea ice, which
enabled GOM formation followed by adsorption onto particles resulting in the unusually high
PBM concentrations over the sea ice (Steffen et al., 2013). In contrast, summertime Arctic GOM
and PBM were very low due possibly to low in situ oxidation of GEM and enhanced physical
scavenging of GOM/PBM as a result of high relative humidity and low visibility (Sommar et al.,



2010).

Higher concentrations of GOM over the *Antarctic* Ocean were proposed by Sprovieri et

al. (2002) to be produced from gas-phase oxidation of GEM by $O_3$, $H_2O_2$, and OH together with
favorable physical conditions such as planetary boundary layer height and perhaps more so by
the latter.  The highest concentrations of GOM corresponding to the lowest concentration falling
below the LOD (0.3 pg m$^{-3}$) during MDEs in summer were associated with the air masses having
a maximum contact with sea ice (coverage >40%) over the South Atlantic Ocean, which was
speculated to contain abundant reactive bromine, Br, released from sea salt associated with sea
ice or sea salt aerosols (Temme et al., 2003b).  Summertime GOM was found to be correlated
with GEM due to in situ oxidation and build-up (Soerensen et al., 2010), and was also observed
to be anti-correlated with GEM due likely to oxidation solely (Temme et al., 2003b; Sprovieri et
al., 2002). Similar to *Arctic* MDEs, air masses during *Antarctic* MDEs appeared to have contact
with sea ice potentially entraining abundant halogen radicals before arriving at the monitoring
location.  Different from the Arctic, summertime GOM concentrations over the Antarctic seemed
to be orders of magnitude larger.
**3. Continental Boundary Layer**

In this section, continental sites are defined as inland sites located in non-polar regions

and exclude locations impacted by the MBL, e.g. coastal sites and the ocean.
***3.1 TGM/GEM***
3.1.1 Concentration Metrics

Field measurements of TGM/GEM at continental sites were conducted mainly in Asia,

Canada, Europe, and USA.  Very few TGM/GEM measurements have been made at inland sites
in the southern hemisphere.  Of all the four regions, the median concentrations of TGM or GEM



were 1.6 ng m$^{-3}$ at remote and rural surface (low elevation) sites, 2.1 ng m$^{-3}$ at urban surface sites,
and 1.7 ng m$^{-3}$ at high elevation sites (**Fig. 2a**). TGM/GEM ranged over 0.1-11.3 ng m$^{-3}$ at
remote sites, 0.2-18.7 ng m$^{-3}$ at rural sites, 0.2-702 ng m$^{-3}$ at urban sites, and 0.6-106 ng m$^{-3}$ at
high elevation sites. Overall these statistics indicate that TGM/GEM at continental urban sites
were higher and had larger variability than rural and remote surface sites and high elevation sites
in the northern hemisphere. By geographical region (**Fig. 2b**), the median TGM/GEM in Asia,
comprising of sites predominantly in China and a few sites in Korea and Japan, were higher by
26-55% than those in Europe, Canada, and USA in this respective order. Although a higher
median TGM/GEM was found in Asia, the maximum single 5-min concentration was recorded in
the USA (324 ng m$^{-3}$, Engle et al., 2010). The 5-min maximum TGM/GEM among the four
regions was the lowest in Europe (23 ng m$^{-3}$, Witt et al., 2010). It is important to note that most
urban sites in the literature are located in North America and Europe, and hence the higher
TGM/GEM at continental urban sites as shown in Figure 2b were predominantly driven by
measurements at those sites (instead of Asian sites). A summary of the mean and the range of
TGM/GEM at individual continental sites can be found in Table S4. Statistics from studies prior
to 2009 are referred to Sprovieri et al. (2010b).

3.1.2 Temporal Variations from Diurnal Cycle to Long-term Trends

3.1.2.1 Diurnal Variation

At *remote* surface locations, the diurnal variation of TGM/GEM is characterized by a

daytime increase reaching a maximum concentration in the afternoon and nighttime decrease
(Manolopoulos et al., 2007; Cheng et al., 2012). The ratio of the daily standard deviation to the
daily mean was 3% in one study (Cheng et al., 2012). Diurnal variations were stronger during
spring than other seasons (Cheng et al., 2012).



884   At *rural* surface and *high elevation* sites, several different diurnal patterns have been

885  reported. The first pattern, similar to remote surface locations, is an early morning minimum,

886  followed by midday to afternoon maximum and decrease at night (Swartzendruber et al., 2006;

887  Yatavelli et al., 2006; Choi et al., 2008, 2013; Fu et al., 2008, 2009, 2010, 2012b; Lyman and

888  Gustin, 2008; Mao et al., 2008; Obrist et al., 2008; Faïn et al., 2009; Sigler et al., 2009; Mazur et

889  al., 2009; Nair et al. 2012; Mao and Talbot, 2012; Eckley et al., 2013; Parsons et al., 2013; Cole

890  et al., 2014; Brown et al., 2015; Zhang et al., 2015). The daytime peak was narrower during

891  winter/fall and broader during spring/summer, similar to the seasonal pattern of daylight hours

892  (Eckley et al., 2013). At elevated sites, the magnitude of this diurnal variation varies with season

893  and location. The diurnal variation was more pronounced during spring at Mt. Gongga, China

894  (Fu et al., 2008, 2009), fall/winter at Storm Peak Laboratory, USA (Faïn et al., 2009), summer at

895  Mt. Changbai, China (Fu et al., 2012b), and winter/spring at Mt. Lulin, Taiwan (Sheu et al.,

896  2010). The diurnal amplitude at Mt. Lulin ranged from 0.34 ng m$^{-3}$ (Fall) to 0.62 ng m$^{-3}$ (winter)

897  or from 17-31%. The second diurnal pattern typically observed is a higher nighttime TGM/GEM

898  than daytime. This tends to occur in Asia and more polluted sites outside of Asia, e.g.

899  abandoned Hg mines and cement plants (Lyman and Gustin, 2008; Wan et al., 2009a;

900  Rothenberg et al., 2010; Li et al., 2011; Nguyen et al., 2011; Fu et al., 2012a; Gratz et al., 2013;

901  Zhang et al., 2013; Cole et al., 2014). In one instance, this diurnal pattern only occurred during

902  winter/fall (Zhang et al., 2013). The third pattern found at rural surface and elevated sites is a

903  weak or lack of diurnal pattern in TGM/GEM (Choi et al., 2008, 2013; Mao et al., 2008; Sigler et

904  al., 2009; Engle et al., 2010; Rothenberg et al., 2010; Mao and Talbot, 2012; Zhang et al., 2013;

905  Han et al., 2014). This pattern was more prominent in specific seasons, e.g. winter (Choi et al.,

906  2008) and spring/summer (Zhang et al., 2013).





At *urban* surface sites, the predominant diurnal pattern is an increase in TGM/GEM
throughout the night that leads to a maximum in the early morning and a decrease in TGM/GEM
in the afternoon (Stamenkovic et al., 2007; Li et al., 2008; Choi et al., 2009; Lyman and Gustin,
2009; Song et al., 2009; Liu et al., 2010; Witt et al., 2010; Nguyen et al., 2011; Nair et al., 2012;
Zhu et al., 2012; Gratz et al., 2013; Kim et al., 2013; Civerolo et al. 2014; Cole et al., 2014; Han
et al., 2014; Lan et al., 2014; Xu et al., 2014; Xu et al., 2015).  The higher nighttime than
daytime pattern was observed during spring, summer, and fall in one study (Civerolo et al. 2014).
The diurnal amplitude was 24% of the daily mean in one study (Song et al., 2009). The diurnal
amplitude can also vary by 9.2-17.8% depending on the location within an urban area (Kim et al.,
2013).  The diurnal amplitude tends to be higher during summer compared to other seasons
(Stamenkovic et al., 2007; Peterson et al. 2009; Civerolo et al. 2014; Lan et al., 2014; Xu et al.,
2014).  Zhu et al. (2012) found a larger diurnal amplitude in the spring (3.7 ng m$^{-3}$) than winter
(0.9 ng m$^{-3}$).  The timing of the TGM/GEM peak also changes between seasons.  TGM decreased
earlier in the day during spring than in other seasons (Xu et al., 2014), while the maximum TGM
occurred later in the morning during spring than in other seasons (Zhu et al., 2012).  Diurnal
variations with daytime maximum and early morning minimum have also been observed at urban
surface sites (Fostier and Michelazzo, 2006; Rothenberg et al., 2010; Witt et al., 2010; Jiang et
al., 2013; Han et al., 2014).  During winter, some studies observed a less pronounced diurnal
variation in TGM/GEM (Choi et al., 2013; Civerolo et al., 2014).
3.1.2.2 Seasonal Variation
The seasonal variation in TGM/GEM at some continental *remote* surface sites can be
characterized by a winter to early-spring maximum and lower summer/fall concentrations
(Manolopoulos et al., 2007; Cheng et al., 2012).  At other *remote* sites, a completely opposite





seasonal pattern was found with higher summer/fall concentrations than winter/spring (Abbott et
al., 2008; Cole et al., 2014). These two seasonal patterns were also reported at *rural surface* and
*elevated* sites. The predominant seasonal TGM/GEM trend at rural surface and elevated sites is
the winter to spring maximum and summer/fall minimum (Zielonka et al., 2005; Yatavelli et al.,
2006; Choi et al., 2008; Fu et al., 2008, 2009, 2010; Mao et al., 2008; Sigler et al., 2009a; Mazur
et al., 2009; Engle et al., 2010; Mao and Talbot, 2012; Nair et al., 2012; Chen et al., 2013; Parson
et al., 2013; Cole et al., 2014; Marumoto et al., 2015). Other studies conducted in *rural* sites and
*elevated* sites found higher TGM/GEM during warm seasons (spring/summer) than in the winter
(Weiss-Penzias et al., 2007; Obrist et al., 2008; Nguyen et al., 2011; Eckley et al., 2013; Zhang
et al., 2013; Zhang et al., 2015). Additional seasonal patterns were observed at *high elevation*
sites, including higher TGM in summer/fall than winter/spring (Fu et al., 2012a) and a spring
maximum and summer minimum in another study (Sheu et al., 2010). The seasonal patterns at
continental *urban* surface sites can by vastly different from each other. They can be summarized
into the following five seasonal patterns: The first pattern is a winter to spring maximum
(Fostier and Michelazzo, 2006; Stamenkovich et al. 2007; Choi et al., 2009; Peterson et al., 2009;
Civerolo et al., 2014; Xu et al., 2015). The second pattern is a summer TGM/GEM maximum
(Xu and Akhtar, 2010; Jiang et al., 2013). The third pattern is higher TGM during both winter
and summer (Xu et al., 2014). The fourth type is higher TGM/GEM during spring/summer than
winter/fall (Liu et al., 2007, 2010; Song et al., 2009; Nair et al., 2012; Zhu et al., 2012; Hall et al.,
2014). The last type reported is an absence of a clear seasonal trend (Kim et al., 2013; Civerolo
et al., 2014; Marumoto et al., 2015). Table 1 summarizes the predominant diurnal and seasonal
patterns observed at rural, urban and high elevation continental sites.
3.1.2.3 Long-term Trends





At *rural* sites across Canada, TGM decreased at a rate of 0.9-3.3% per year between 1995
and 2011.  Depending on the location, the trend was determined using 5-15 years of TGM data
(Cole et al., 2014).  A GEM decrease of 0.056 ng m$^{-3}$ yr$^{-1}$ from 2005-2010 was found at an
*elevated* site in New Hampshire (Mao and Talbot, 2012). Widespread declines in GEM across
North America between 1997 and 2007 have also been reported (Weiss-Penzias et al., 2016);
however, the trends were not determined separately for rural and urban sites.  No significant
trends in TGM were found at *urban/industrial* sites in the UK from 2003-2013 (Brown et al.,
2015) and at another urban site in Seoul, Korea from 2004-2011 (Kim et al., 2013).  However, a
short-term annual TGM decrease from 2.0 to 1.7 ng m$^{-3}$ was recorded at an urban site in Windsor,
Canada from 2007-2009 (Xu et al., 2014).  At a chlor-alkali site in the UK, TGM declined by
1.36 ± 0.43 ng m$^{-3}$ yr$^{-1}$ from 2003-2012 (Brown et al., 2015).  Weigelt et al. (2015) determined
annual TGM trends for different air masses arriving at Mace Head, Ireland between 1996 and
2013.  Specifically for continental airflows, TGM decreased by 0.0240 ± 0.0025 ng m$^{-3}$ yr$^{-1}$ for
polluted air masses from Europe, which was a slightly faster decline compared to marine
airflows from the North Atlantic Ocean (-0.0209 ± 0.0019 ng m$^{-3}$ yr$^{-1}$) and southern hemisphere
(-0.0161 ± 0.0020 ng m$^{-3}$ yr$^{-1}$).  In certain months, the TGM decreases associated with local and
European airflows (0.047-0.051 ng m$^{-3}$ yr$^{-1}$) were greater than other months (Weigelt et al.,

2015).

3.1.3 Mechanisms Driving the Observed Temporal Variabilities
3.1.3.1 Diurnal Variation of TGM/GEM
TGM/GEM was higher during daytime than nighttime and often declined to a minimum
in the early morning at remote, rural, high elevation, and some urban surface sites (Table 1).
One of the mechanisms driving this diurnal pattern involves meteorological parameters, such as



temperature, the increase of which enhances TGM/GEM volatilization (Manolopoulos et al.,
2007; Mao et al., 2012; Jiang et al., 2013; Han et al., 2014).  Surface emissions of TGM can
occur during daytime from soil and snow as temperature and solar radiation increases (Mao et al.,
2012; Cole et al. 2014).  Solar radiation minimizes the activation energy required for Hg
emissions (Zhu et al., 2012) and increases Hg photoreduction in soil and snow (Steffen et al.,
2008; Zhu et al., 2012; Hall et al., 2014; Xu et al., 2014; Xu et al., 2015).  This process is
especially relevant at sites with elevated Hg in soil (Lyman and Gustin, 2008; Brown et al., 2015)
because of a larger flux gradient.  Dry deposition of GEM in the night might also play a role
since deposition was typically observed in nighttime in contrast to emission during daytime
(Zhang et al., 2009).  Fog or dew formation occurring in the late summer was believed to have
caused GEM depletion in the early morning hours by capturing GEM in fog or dew water
(Manolopoulos et al., 2007; Mao and Talbot, 2012).  Another driving mechanism of this
TGM/GEM diurnal pattern is the change in the boundary layer mixing height.  Lower
TGM/GEM during nighttime is due to TGM/GEM deposition as the nocturnal inversion layer
forms.  In the morning, the nocturnal inversion breaks down and mixes with TGM/GEM-rich air
in the residual layer and subsequently leads to increasing TGM/GEM during the day (Yatavelli et
al., 2006; Mao et al., 2008; Mazur et al., 2009; Mao and Talbot, 2012; Nair et al. 2012; Choi et
al., 2008, 2013; Jiang et al., 2013; Cole et al., 2014).  At elevated sites, there is a transition from
the sampling of boundary layer during daytime to free troposphere air at night which is driven by
mountain/valley atmospheric patterns (Obrist et al., 2008).  During daytime, mountain breezes
causes moist air to ascend from the surface to higher altitudes carrying with it GEM from the
boundary layer (Swartzendruber et al., 2006; Obrist et al., 2008; Fu et al., 2010, 2012b; Zhang et
al., 2015).  At night, drier free troposphere air impacts the elevated site leading to lower GEM





and water vapor and higher GOM and ozone (Obrist et al., 2008). The shift in prevailing wind
directions also contributed to this higher daytime TGM diurnal pattern in one study (Fu et al.,
2008, 2009). A lack of diurnal variability was also reported at some rural surface locations,
although the driving mechanism is not quite clear. At an elevated site, the sampling of air above
the nocturnal boundary layer and lack of anthropogenic sources or GEM oxidants near the site
led to constant GEM during most of the time except in the summer (Mao et al., 2008; Sigler et al.,
2009a; Mao and Talbot, 2012). Thus this differs from other mountain sites, which were affected
by surface emissions and local/regional transport of GEM from the boundary layer during
daytime.

At most urban sites and some elevated and polluted rural sites, the nighttime TGM

concentrations were higher than daytime, and the maximum concentration typically occurred in
the early morning before sunrise (Table 1). This type of diurnal variation is driven by nighttime
accumulation of TGM/GEM near the surface due to a shallow nocturnal boundary layer and
dilution during the day initiated by convective mixing with cleaner air aloft as the mixing layer
increases (Stamenkovic et al., 2007; Li et al., 2008; Lyman and Gustin, 2008, 2009; Choi et al.,
2009; Wan et al., 2009a; Rothenberg et al., 2010; Witt et al., 2010; Li et al., 2011; Nguyen et al.,
2011; Fu et al., 2012a; Nair et al., 2012; Zhu et al., 2012; Gratz et al., 2013; Kim et al., 2013;
Zhang et al., 2013; Cole et al., 2014; Lan et al., 2014; Xu et al., 2014). The shallow nocturnal
boundary layer is often associated with high TGM coinciding with low wind speeds at night (Li
et al., 2008; Fu et al., 2012a; Lan et al., 2014). Increases in nighttime concentrations can also be
driven by nighttime sources, such as emissions from mercury mining regions (Lyman and Gustin,
2008) and local emissions occurring at night (Song et al., 2009; Wan et al., 2009a; Rothenberg et
al., 2010; Gratz et al., 2013; Kim et al., 2013). One study suggested that the evening TGM



maximum was attributed to coal combustion and biofuel burning (e.g. wood/leaves) for cooking
and coincided with winds travelling over residential areas in China (Wan et al., 2009a). The
morning TGM/GEM maximum at a rural site after sunrise may be attributed to foliar emissions
(Nguyen et al., 2011). At urban surface sites, studies suggested the driving mechanisms for the
morning maximum were surface emissions (Zhu et al., 2012; Hall et al., 2014; Xu et al., 2014;
Xu et al., 2015), volatilization of Hg from dew (Zhu et al., 2012), and vehicular traffic emissions
evident by correlations between TGM/GEM and CO and $NO_x$ (Zhu et al., 2012; Xu et al., 2015).
However, there is little research suggesting significant amounts of Hg from vehicular emissions.
The lower TGM/GEM observed in the afternoon was driven by GEM oxidation (Stamenkovic et
al., 2007; Choi et al., 2009; Lyman and Gustin, 2009; Li et al., 2011; Nguyen et al., 2011; Kim et
al., 2013; Zhang et al., 2013; Xu et al., 2014; Xu et al., 2015). Sea breeze also affected the
diurnal pattern at an inland urban site (Lan et al., 2014). Sea breezes transported cleaner marine
air 70 km inland to Houston, Texas leading to lower TGM in the afternoon on most days, similar
to coastal sites (Cole et al., 2014). However on some days, the southerly sea breeze was
intersected by northerly flows which led to a period of stagnant air. The lack of pollutant
dispersion led to an increase in TGM (Lan et al., 2014).

Many studies conducted in urban areas found a larger diurnal amplitude during summer

than other seasons. The major driving mechanism for this larger amplitude originated from
higher solar radiation and temperature, which increased the boundary layer mixing height in the
summer (Civerolo et al., 2014; Xu et al., 2014). Higher solar radiation during summer also
increased photochemical reactions, like GEM oxidation. The larger diurnal variation was also
attributed to increases in uptake and re-emissions by vegetation and power plant emissions from
air conditioner use during summer nights (Xu et al., 2014). The shift in the timing of the



TGM/GEM maximum varied with season at some urban sites as previously mentioned.  During
spring in Windsor, Canada, the decrease in TGM earlier in the afternoon was thought to be due
to increase photochemical processes resulting from higher solar radiation and lower GEM
emissions due to less vegetation coverage in the spring (Xu et al., 2014).  In Nanjing, China, the
peak concentration occurring later in the morning during spring was driven by prolonged
sunlight hours (Zhu et al., 2012).

Site characteristics may have different impacts on the diurnal variation.  During nighttime,

GEM at an urban site was significantly higher than a rural site suggesting higher GEM fluxes
from buildings and pavement than vegetation and soil (Liu et al., 2010). The diurnal amplitude at
an urban site was greater than a suburban site in one study; however, the reason is not known
(Civerolo et al., 2014).  In the same study, nighttime GEM was 25-30% higher than daytime for
the urban site close to the Atlantic Ocean, whereas the GEM difference between night and day
was only 10% at an inland suburban site (Civerolo et al., 2014).  The study suggested that the
higher halogen concentrations in marine environments increased GEM oxidation and
subsequently, the loss of GEM in the afternoon leading to larger diurnal variation.  At a different
coastal-urban location, nighttime GEM was only slightly higher than daytime because of the
cleaner air transported from the marine environment (Nguyen et al., 2011).  These studies
suggested that MBL influence could lead to very different diurnal patterns.  Sites continuously
impacted by Hg point sources likely contributed to the large short-term fluctuations in the diurnal
patterns at some urban sites (Rutter et al., 2008; Engle et al., 2010; Witt et al., 2010).

3.1.3.2 Seasonal Variation of TGM/GEM

The seasonal variation exhibiting a winter to spring maximum in remote, rural, urban and

high elevation environments (Table 1) was suggested to be driven by multiple mechanisms,



including anthropogenic emissions for winter heating (coal and wood combustion), reduced
atmospheric mixing, decreased GEM oxidation, less scavenging, and emissions from soil,
vegetation, and melting snow in the spring (Stamenkovic et al., 2007; Choi et al., 2008; Mao et
al., 2008; Sigler et al., 2009a; Peterson et al., 2009; Wan et al., 2009a; Cheng et al., 2012; Mao
and Talbot, 2012; Civerolo et al., 2014; Cole et al., 2014; Xu et al., 2015). The lower
TGM/GEM during summer has been attributed to increased GEM oxidation, uptake by
vegetation, and higher wet deposition of GOM (Yatavelli et al., 2006; Fu et al., 2008, 2009;
Engle et al., 2010; Xu et al., 2015). While these were the predominant driving mechanisms of
the seasonal variations in the northern hemisphere, the seasonal patterns could also be influenced
by changes in the prevailing wind patterns (Fostier and Michelazzo, 2006; Fu et al., 2010, 2015;
Sheu et al., 2010; Chen et al., 2013). The impact of combustion emissions from winter heating
was ruled out at a subtropical site in the Pearl River Delta region of China (Chen et al., 2013).
Chen et al. (2013) attributed the elevated TGM in the spring to monsoons, which advected
southerly marine air masses during summer and northeasterly winds from Siberia during winter.
The transition from cold dry air to warm moist air often led to strong temperature inversion and
haze in the spring, which in turn inhibits pollutant dispersion. In Brazil, higher TGM in
December than May was driven by airflow travelling over high traffic areas in December and
then a switch to airflows travelling over vegetation in May (Fostier and Michelazzo, 2006).
Summer and spring maxima in TGM/GEM have also been found at remote, rural, and
urban atmospheres. This pattern is predominantly driven by meteorology. Higher solar radiation
and temperature during summer increased GEM emissions from Hg contaminated soil (Zhu et al.,
2012; Eckley et al., 2013), from vegetation at a forested agricultural site (Nguyen et al., 2011),
and from urban surfaces such as soil and pavement in Windsor, Canada (Xu and Akhtar, 2010).



One study attributed the large difference in the mean TGM between summer and winter (4.4 ng
$m^{-3}$) and frequent elevated TGM events (>12 ng $m^{-3}$) during summer to surface to air fluxes from
Hg contaminated soil in Nanjing (Zhu et al., 2012). This was further supported by TGM
correlation with temperature and solar radiation and weak correlation with CO during summer, in
which the latter is a strong tracer of anthropogenic emissions. In addition to GEM emissions
increasing in warm seasons, higher TGM during summer was attributed to lower wind speeds
which prevent pollutant dilution, and increase power plant emissions resulting from higher
energy consumption for cooling (Xu et al., 2014). Seasonal change in the prevailing wind
direction was another mechanism contributing to this seasonal TGM pattern in China (Fu et al.,
2012a,b, 2015; Zhang et al., 2013; Hall et al., 2014). During spring, summer and fall, the
prevailing winds from the southwest transported Hg from polluted regions of China to Beijing,
whereas cleaner air from the northwest arrived in Beijing during winter (Zhang et al., 2013).
The summer monsoon advected biomass burning and industrial emissions from the Pearl River
Delta, which also contributed to the summer TGM maximum in Nanjing, China (Hall et al., 2014)
in addition to soil emissions discussed earlier.
3.1.3.3 Long-term Trends of TGM/GEM
Long-term trends of TGM/GEM over continental regions indicated a declining trend at
some sites and no significant trend at others, particularly at urban sites. Previous studies partly
attributed the long-term TGM trends to anthropogenic Hg emissions reductions. There has been
a 60-70% decrease in anthropogenic Hg emissions from USA and Canada; however only up to
15% of those emissions reductions impacted TGM at Canadian sites (Cole et al., 2014). The
more rapid decline in TGM measured at Mace Head, Ireland for local and European air masses
compared to marine air masses was thought to be driven by Hg emissions reductions in Europe



(Weigelt et al., 2015). The baseline TGM at Mace Head decreased at a larger rate in November
than other months suggesting that it is related to lower Hg emissions from residential heating in
Europe. The 21% decline in TGM from 2006-2012 in urban/industrial areas of the UK was also
consistent with the 0.21 Mg yr$^{-1}$ (24%) reduction in Hg emissions from the UK, even though the
TGM trend from the 2003-2013 period was not statistically significant (Brown et al., 2015). In
Seoul, Korea, no significant trend in TGM was found from 2004-2011, consistent with the slight
decrease (1%) in coal consumption in Seoul over the same time frame (Kim et al., 2013). While
TGM/GEM trends appear to be aligned with local/regional Hg emission trends, a discrepancy
exists when the trend was compared to the increasing global anthropogenic Hg emissions
(Sprovieri et al., 2010b; Ebinghaus et al., 2011; Cole et al., 2014). Alternative reasons for the
decline in TGM could be due to faster cycling of Hg as ozone and other oxidants have been
increasing or lower emissions of previously-deposited Hg (Sprovieri et al., 2010b; Ebinghaus et
al., 2011). Modeling studies indicate global Hg emissions inventory have not accounted for the
changes in Hg speciation emission profiles from coal combustion and reduced emissions from
products containing Hg (Zhang et al., 2016).

***3.2 GOM and PBM***
3.2.1 Concentration Metrics
The highest median GOM and PBM were found at high elevation sites, while the lowest
concentrations were found at rural surface sites. The median GOM from all locations were 12.1
pg m$^{-3}$ at elevated sites, 9.9 pg m$^{-3}$ at urban sites, 3.8 pg m$^{-3}$ at remote sites, and 2.8 pg m$^{-3}$ at
rural sites (Fig. 2a), and correspondingly the median PBM concentration was 11.0, 10.0, 6.9, and
4.6 pg m$^{-3}$. The variabilities in GOM and PBM were greatest at urban locations. 2-3 hour GOM



concentrations ranged from <LOD-880 pg m$^{-3}$ at elevated sites, <LOD-8160 pg m$^{-3}$ at urban sites,
<LOD-224 pg m$^{-3}$ at remote sites, and <LOD-462 pg m$^{-3}$ at rural sites (see individual site
statistics in Table S5). The large variability in GOM was also observed in PBM. 2-3 hour PBM
concentrations ranged from <LOD-1001 pg m$^{-3}$ at elevated sites, <LOD-11600 pg m$^{-3}$ at urban
sites, <LOD-404 pg m$^{-3}$ at remote sites, and <LOD-205 pg m$^{-3}$ at rural sites (Table S6). The
large variability at remote sites is due to the presence of coal-fired power plants within 100 km
of one of the sites. By geographical region, the median GOM in Asia was a factor of 1.4-5.1
higher than those in Canada and USA (Fig. 2b). Similarly, the median PBM in Asia was 1.8-8.1
times higher than those in Canada, Europe and USA. This is potentially because one-third of the
elevated sites were in China. The GOM and PBM maxima of 8160 pg m$^{-3}$ and 11600 pg m$^{-3}$,
respectively, were both observed at an urban site in Illinois, USA (Engle et al., 2010; Table S5
and S6).

3.2.2 Temporal Variations from Diurnal Cycle to Seasonal Trends

3.2.1.1 Diurnal Variation

The predominant diurnal pattern of GOM at remote, rural, urban, and elevated sites is an

increase in the morning leading to a maximum sometime between midday to late afternoon and
eventually decreasing at night (Yatavelli et al., 2006; Manolopoulos et al. 2007; Abbott et al.,
2008; Lyman and Gustin, 2008; Faïn et al., 2009; Rothenberg et al., 2010; Cheng et al., 2012; Fu
et al., 2012a; Nair et al., 2012; Eckley et al., 2013; Gratz et al., 2013; Cole et al., 2014; Civerolo
et al., 2014; Marumoto et al. 2015; Zhang et al., 2015). At a *remote* Canadian site, the ratio of
the standard deviation to the daily mean of GOM for this type of diurnal pattern was 52%
(Cheng et al., 2012). The GOM diurnal amplitudes varied by 35-180% across Canadian *rural*
sites (Cole et al., 2014). In *urban* areas, the daytime GOM can be 2-3 folds higher than



nighttime during spring and summer (Civerolo et al., 2014). The diurnal amplitude was larger
during spring/summer than fall/winter (Peterson et al., 2009; Cheng et al., 2012; Mao and Talbot,
2012; Choi et al., 2013) and at urban sites compared to rural sites (Nair et al., 2012; Cheng et al.,

2014).

In addition to this diurnal pattern, GOM was elevated throughout the day and night at a

higher latitude remote site (Cole et al., 2014). A weak diurnal variation was also observed at
*rural* sites (Cobbett and Van Heyst, 2007; Choi et al., 2008; Rothenberg et al., 2010), urban sites
(Rothenberg et al., 2010; Civerolo et al., 2014; Xu et al., 2015), and an elevated site (Sigler et al.,
2009; Mao and Talbot, 2012; Mao et al., 2012). Unlike rural sites, diurnal patterns at *urban* and
*elevated* sites can appear opposite to the higher daytime diurnal pattern. Late evening increases
in GOM were observed at some urban sites (Lynam and Keeler, 2005; Song et al., 2009; Gratz et
al., 2013), resulting in a lower diurnal amplitude of 14% in one study (Song et al., 2009). Some
high altitude sites observed higher GOM (average: 18-60 pg m$^{-3}$) between midnight and early
morning (Swartzendruber et al., 2006; Sheu et al., 2010). By comparison, the average daytime
GOM at these sites were 9.2-39 pg m$^{-3}$.

No predominant diurnal pattern was found for PBM, which was mostly measured using

the Tekran speciation unit (2537-1135-1130). At *rural* sites, the types of diurnal patterns include,
daytime/afternoon peak (Yatavelli et al., 2006; Choi et al., 2008; Rothenberg et al. 2010; Cole et
al., 2014), increasing during daytime leading to a nighttime peak (Nair et al., 2012; Zhang et al.,
2013), or lack of variation (Cobbett and Van Heyst, 2007; Choi et al., 2008; Rothenberg et al.,
2010; Cole et al., 2014). These three patterns were also found at *urban* sites. For the higher
daytime pattern, daytime PBM can be 1.5-2 times higher than nighttime during spring (Civerolo
et al., 2014). In comparison, the diurnal amplitude was only 14% of the daily mean at an *urban*



site (Song et al., 2009).  At *high elevation* sites, higher daytime/afternoon (Fu et al., 2012a) and a
lack of variation were observed (Sheu et al., 2010; Zhang et al., 2015).

3.2.1.2 Seasonal Variation

No predominant seasonal pattern in GOM was found at remote, rural, urban, and elevated

sites.  At *remote* sites, some studies observed a winter to early-spring maximum and lower
concentrations during summer/fall (Manolopoulos et al., 2007; Cheng et al., 2012), whereas
higher summer/fall than winter/spring concentrations were also reported (Abbott et al., 2008).  In
*rural* areas, the maximum concentration can occur in different seasons.  The maximum GOM
was found in spring and minimum in the fall at most Canadian sites (Cole et al., 2014), except
for a summer maximum observed at one Canadian rural site (Eckley et al., 2013).  The seasonal
maxima in GOM at other *rural* sites can also occur during spring/fall (Nair et al., 2012),
winter/summer (Choi et al., 2008), and summer/fall (Zhang et al., 2013).  Han et al. (2014) did
not observe a seasonal pattern.  At *urban* sites, the maximum GOM typically occurred in warmer
seasons, e.g. spring or summer (Song et al., 2009; Liu et al., 2010; Choi et al., 2013; Wang et al.,
2013; Gratz et al., 2013; Civerolo et al., 2014; Han et al., 2014; Marumoto et al., 2015; Xu et al.,
2015).  An exception to this seasonal pattern is the higher fall and winter concentrations in
northern Mississippi, USA (Jiang et al., 2013).  The maximum GOM was also reported in
different seasons at *elevated* sites.  The maximum GOM was found sometime between fall and
spring at mountain sites in China (Wan et al., 2009b; Sheu et al., 2010; Fu et al., 2012a; Zhang et
al., 2015) and an elevated site in the U.S. (Sigler et al., 2009a).  This contrasts the summer
maximum of reactive mercury (GOM+PBM) at three elevated western U.S. sites (Weiss-Penzias
et al., 2015).



Higher PBM and total particulate Hg (TPM) during colder seasons than summer is a
highly ubiquitous trend for remote, rural, urban, and elevated sites (Zielonka et al, 2005; Choi et
al., 2008; Wan et al., 2009b; Liu et al., 2010; Kim et al., 2012; Gratz et al., 2013; Beldowska et
al., 2012; Marumoto et al., 2015; Schleicher et al., 2015; Zhang et al., 2015).  In one study, the
TPM fraction was 20.2% of TGM during winter and 6.3% during summer (Zielonka et al, 2005).
Beldowska et al. (2012) also observed a larger fraction of TPM during the heating season (0.1-
12%) than summer (0.1-3%).  However, increases in PBM also occurred during summer in a few
studies (Song et al., 2009; Huang et al., 2010; Cheng et al., 2012).
3.2.3 Mechanisms Driving the Observed Temporal Variabilities
3.2.3.1 Diurnal Variations of GOM and PBM
The widespread observation of a midday to late afternoon peak in GOM at continental
sites (Table 1) often coincided with meteorological parameters, such as solar radiation and
temperature, and/or ozone (Yatavelli et al., 2006; Abbott et al., 2008; Wan et al., 2009a; Weiss-
Penzias et al., 2009; Nair et al., 2012; Mao et al., 2012; Gratz et al., 2013; Zhang et al., 2013;
Civerolo et al., 2014; Cole et al., 2014; Marumoto et al., 2015).  At *high elevation* sites, GOM
was also inversely correlated with relative humidity, water vapor, or dew point temperature
(Swartzendruber et al., 2006; Lyman and Gustin, 2008, 2009; Weiss-Penzias et al., 2009), and in
some cases GOM was not correlated with ozone (Lyman and Gustin, 2009; Peterson et al., 2009;
Xu et al., 2015).  These diurnal trends infer daytime *in-situ* photochemical production of GOM
or entrainment of GOM from the free troposphere due to convective mixing.  Increases in GOM
during daytime at a rural site was attributed to local transport from urban areas as indicated by
similarities in diurnal patterns between GOM, $SO_2$, and $O_3$ and a delay in the timing of the GOM
maximum likely resulting from emissions transport (Rothenberg et al., 2010).  Short-term



fluctuations in the diurnal pattern of GOM also suggested the influence of point sources (Rutter
et al., 2008; Engle et al., 2010).  Dry deposition and scavenging of GOM by dew played a role in
decreasing GOM during nighttime (Liu et al., 2007; Wan et al., 2009b; Weiss-Penzias et al.,
2009; Nair et al., 2012; Choi et al., 2013; Civerolo et al., 2014).  The stronger diurnal amplitude
during the spring/summer coincided with stronger correlations between GOM, solar radiation,
temperature and $O_3$ (Yatavelli et al., 2006; Mao et al., 2012; Gratz et al., 2013; Zhang et al.,
2013), which suggested that increased photochemical processes led to higher GOM.  Large
diurnal variation during summer was also potentially driven by high pressure, drier and cloud-
free conditions that are conducive to the buildup of GOM in the free troposphere (Lyman and
Gustin, 2009).

Nighttime increases in GOM seen exclusively at *urban* and *elevated* sites (Table 1)

appeared to be driven by anthropogenic emissions and the free troposphere. Nocturnal emissions
and local/regional transport within the boundary layer (Lynam and Keeler, 2005; Song et al.,
2009) and reduced vertical mixing in the stable nocturnal boundary layer led to higher GOM at
night in *urban* areas (Gratz et al., 2013).  At *high elevation* sites, katabatic winds entrained GOM
from the free troposphere.  In one study, GOM from the free troposphere was believed to
originate from *in-situ* photochemical processes due to a strong inverse GEM-GOM correlation
and a GOM/GEM slope near unity during an elevated GOM episode (Swartzendruber et al.,
2006).  While an anti-correlation between GEM and GOM was also found at another elevated
site, Sheu et al. (2010) did not observe a complete photochemical conversion of GEM to GOM.
The difference between these two *elevated* sites suggests different sources of GOM in the free
troposphere.  Timonen et al. (2013) found that in one type of free troposphere air mass, GEM
oxidation occurred in anthropogenic plumes transported from Asia to Mt. Bachelor Observatory,





USA and converted 20% of the GEM to GOM.  A second type of air mass travelling over the
Pacific Ocean resulted in 100% GEM conversion to GOM likely because of GEM oxidation by
bromine.

The driving mechanisms behind the diurnal pattern of PBM were better explored for

*urban* sites than other site categories.  Frequent spikes in hourly concentrations during daytime
were attributed to point sources (Rutter et al., 2008; Civerolo et al., 2014).  At a valley *urban* site,
higher PBM and GEM during daytime suggested similar emission sources from Hg enriched
areas (Lyman and Gustin, 2009).  Higher PBM during daytime in the summer could also be
initiated by photochemical production of GOM followed by absorption on secondary organic
aerosols (Choi et al., 2013).  Diurnal patterns exhibiting nighttime increases in PBM in urban
areas could be due to multiple mechanisms and sources, such as nocturnal emissions and
local/regional transport within the boundary layer (Song et al., 2009), reduced vertical mixing in
the stable nocturnal boundary layer (Gratz et al., 2013; Xu et al., 2015), vehicular emissions in
China (Xu et al., 2015), and nighttime street food vending in Beijing (Schleicher et al. 2015).

3.2.3.2 Seasonal Variations of GOM and PBM

The seasonal variation characterized by higher GOM in the warm seasons (Table 1) is

primarily driven by photochemical production due to increased solar radiation, $O_3$, and likely
other atmospheric oxidants (Liu et al., 2010; Choi et al., 2013; Civerolo et al., 2014; Xu et al.,
2015).  Alternative reasons could be attributed to anthropogenic emissions leading to higher
GOM in the summer at urban sites (Song et al., 2009; Gratz et al., 2013).  Atmospheric mercury
depletion events occurring at higher latitude continental sites led to higher GOM during spring
(Cole et al., 2014).  Free troposphere transport was a major driving mechanism for higher
reactive Hg at three high elevation western U.S. sites (Weiss-Penzias et al., 2015).  In one study,



increases in GOM during fall and winter coincided with increases in traffic at a university
campus when classes were in session (Jiang et al., 2013). At *elevated* sites in China, the
occurrence of higher GOM between fall and spring were attributed to coal and biofuel burning
(Wan et al., 2009b) and changes in the prevailing winds that advected GOM from polluted
regions (Fu et al., 2012a; Zhang et al., 2015). Lower GOM during summer was due to wet
deposition (Wan et al., 2009b; Sheu et al., 2010).
Several mechanisms contribute to the increase in PBM or TPM during colder seasons
(Table 1) including, local/regional coal combustion and wood burning emissions, lower mixing
height, less oxidation, and increased gas-particle partitioning (Song et al., 2009; Xiu et al., 2009;
Liu et al., 2010; Cheng et al., 2012; Fu et al., 2012a; Kim et al., 2012; Choi et al., 2013; Gratz et
al., 2013; Wang et al., 2013; Civerolo et al., 2014; Cole et al., 2014; Schleicher et al., 2015; Xu
et al., 2015). Oxidized Hg tended to partition to particles during colder seasons because of lower
temperatures (Rutter et al., 2007), higher relative humidity (Kim et al., 2012), and reduced
volatilization of gaseous Hg (Choi et al., 2013). Similar to GOM, decreases in PBM during
summer at many sites in China were due to wet deposition (Wan et al., 2009b; Schleicher et al.,
2015; Xu et al., 2015; Zhang et al., 2015) and a shift to cleaner marine airflows during summer
(Kim et al., 2012). Higher PBM during warm seasons may be driven by forest fire emissions
(Eckley et al., 2013) and increased $PM_{2.5}$ available for GOM absorption at urban sites (Song et
al., 2009; Schleicher et al., 2015).
**4. Latitudinal Variation**
There are a few shipboard and airborne studies that surveyed latitudinal variation of
TGM/GEM (Slemr et al., 1981, 1985, 1995; Slemr and Langer, 1992; Fitzgerald et al., 1984;
Lamborg et al., 1999; Temme et al., 2003a; Aspmo et al., 2006; Soerensen et al., 2010). Overall,



the composite latitudinal distribution from studies of the past three decades showed that
TGM/GEM concentrations over the ocean surface decreased from NH to SH (**Figs. 3 & 4**), with
the highest concentrations (~3.5 ng m$^{-3}$) in northern hemispheric midlatitudes and the lowest in
southern hemispheric latitudes (~0.9 ng m$^{-3}$). Slemr et al. (1981, 1985) found that the
concentrations remained relatively constant (1.4 – 1.6 ng m$^{-3}$) in NH, dropped rapidly once the
ship passed the ITCZ at about 12°N – 13°N latitude, with natural variability of 16%, and varied
over 1.0 – 1.2 ng m$^{-3}$ in the South Atlantic. In addition, Temme et al. (2003a) found higher
variability of TGM in NH (21% vs. 8% in the southern hemisphere) suggesting the majority of
Hg emissions were located in NH, refuting the hypothesis of large oceanic sources of Hg by
previous work (e.g., Mason et al., 1994).

Bagnato et al. (2013) compiled a latitudinal distribution of GEM using measurement data

from a number of shipboard measurement studies spanning the time period of 1980 – 2012 (**Fig.**
**3**) and showed a small but discernible inter-hemispheric gradient in GEM resulting from greater
emissions of Hg to the atmosphere in the more industrialized NH. However, caution should be
taken that global anthropogenic emissions had decreased significantly over that time period, and
the trend in natural emissions (reemissions) was unknown (Slemr et al., 2010).

Tropospheric airborne measurements from INTEX-B (Talbot et al., 2007, 2008) and

ARCTAS (Mao et al., 2010), spanning near the surface to 12 km altitude, suggested on average
~50 ppqv (~0.5 ng m$^{-3}$) increases in GEM concentrations from lower latitudes (~20 – 30 °N) to
higher (60 – 90°N) latitudes in spring, while negligible latitudinal variation in summer (Fig. 2).
There seemed to be distinct seasonal variation in concentrations and latitudinal gradient. It was
speculated that smaller latitudinal gradient of temperature in summer likely enhanced meridional
circulation resulting in smaller latitudinal variation in GEM concentration in the troposphere.



A small gradient was measured in atmospheric GEM concentrations over the Pacific from
1.32 ng m$^{-3}$ in 14 – 20°N latitudes to 1.15 ng m$^{-3}$ in 1-15°S latitudes in October 2011 (Soerensen
et al., 2014).  Atmospheric GEM elevated in the northern part of the ITCZ was temporarily
influenced by the northeastern trade wind that enhanced oceanic evasion, consistent with the
largest evasion flux in that region.
**5. Altitude Variation**
Airborne measurements of TGM, GEM, and/or GOM have been conducted since 1977
(Seiler et al., 1980) extending from near the surface to ~12 km altitude at several geographic
locations (Table S7; references therein).  Tropospheric GEM, GOM, and PBM concentrations
have not thus far been mapped out globally, and a general understanding is lacking on the
mechanisms driving the distributions.
Early studies had conflicting reports of the vertical gradient of TGM.  Seiler et al. (1980)
reported vertical and hemispheric gradient in TGM, 2.7 ng m$^{-3}$ at 1-3 km and 1.5 ng m$^{-3}$ at 8 km
altitude over the Pacific west of San Francisco, and at 8 km altitude 1.45± 0.22 ng m$^{-3}$ and
1.08±0.36 ng m$^{-3}$ in the northern and southern hemisphere, respectively.  They attributed higher
concentrations of TGM (2.4 – 2.7 ng m$^{-3}$) in the ITCZ to convective transport.  In contrast, Slemr
et al. (1985) suggested vertically well-mixed TGM in the troposphere based on their average
concentration of TGM (2.24±0.51 ng m$^{-3}$) at 6 – 8 km altitude over central Europe being close to
their NH midlatitude shipboard measurements conducted in the same study. A similar range of
upper tropospheric GEM was reported by Ebinghaus et al. (2007) and elevated GEM
concentrations in biomass burning plumes from the same study suggested biomass burning
representing a major mercury source.



*Vertical profiles* showing nearly constant, slightly decreasing GEM with altitude were

measured in more studies (Banic et al., 2003; Radke et al., 2007; Talbot et al., 2007, 2008; Mao

et al., 2010). Moreover, *seasonal variation* was observed in flights from surface to 7 km over

Canada with ~1.5 ng m$^{-3}$ in summer, 1.7 ng m$^{-3}$ in winter, 1.7 ng m$^{-3}$ >1 km altitude and 1.2 ng

m$^{-3}$ below 1 km due to widespread MDEs over the sea ice in the springtime Arctic (Banic et al.,

2003). During ARCTAS, Mao at el. (2010) found that the vertical extent of springtime Arctic

MDEs varied from meters to 1 km depending on the thickness of the surface inversion layer.

Over the West Pacific near the Californian coastline, it was found that GEM decreased

distinctly with altitude above relatively constant concentrations from the surface to ~ 3–4 km

altitude, associated with a marked decrease in GEM under stratospherically influenced

conditions, and it was hypothesized that the upper troposphere/lower stratosphere (UTLS) was a

Hg sink region (Radke et al., 2007). Depleted GEM in stratospheric air was observed repeatedly

by Talbot et al. (2007, 2008) during INTEX-B flights at ~ 12 km altitude over the Pacific

Northwest, near Honolulu, HI and Anchorage, AK, USA as well as over the western US near the

Pacific coast. Coupled with Murphy et al. (1998, 2006)'s findings of enrichment of PBM in

lower stratospheric aerosols, Talbot et al. (2007) hypothesized that GEM depletion was caused

by fast oxidation of GEM by abundant halogen radicals and $O_3$ in that region and estimated a

lifetime of 2 and 0.5 days for 100 ppqv GEM oxidized by $O_3$ and Br, respectively. Talbot et al.

(2007) suggested that stratospheric intrusion could be a source of tropospheric Hg if PBM was to

be transformed back to gaseous Hg.

In the atmosphere of East Asia, Friedli et al. (2004) was the first to report Hg GEM

concentrations from sea level to ~7 km altitude under the influence of continental export from

East China, showing concentrations at all altitudes higher than the global background, with the



largest 6.3 ng m$^{-3}$ in an industrial plume mostly from coal combustion and at times from other
sources including dust storms, biomass burning, and volcanic eruption.

Unlike measurements from the studies aforementioned, Swartzendruber et al. (2008)

found that layer-averaged GEM concentrations increasing with altitude from 1.30±0.084 ng m$^{-3}$
in 0 – 0.5 km altitude to 1.52±0.182 ng m$^{-3}$ in the highest layer 5.5 – 6.5 km altitude over the
Pacific Northwest over 13 April – 16 May 2006.  The higher GEM concentrations above 2.5 km
were associated with long range transport of Asian pollution based on the positive GEM-CO
correlation and back trajectories.

Upper air GOM concentrations were first measured in spring by Lindberg et al. (2002) at

1000 m (exterior to the boundary layer) and 100 m altitude (within the boundary layer)
immediately northeast of Point Barrow. Six aircraft surveys consistently showed that GOM
concentrations decreased from an average 70 to 20 to 2 pg m$^{-3}$ from 5 to 100 to 1000 m altitude,
supporting the hypothesis that the Hg oxidation reactions occurred in the near-surface boundary
layer driven by halogen compounds derived from sea-salt aerosols.

In recent years, more studies found that higher GOM concentrations in higher altitudes

were attributed to lack of depositional loss, lower temperature, and/or more abundant Br radicals
(Sillman et al., 2007; Lyman and Jaffe, 2011; Brooks et al., 2014; Gratz et al., 2015; Shah et al.,
2016).  Sillman et al. (2007) reported GOM concentrations measured in Florida varying from 10
to 230 pg m$^{-3}$ increasing with height.  They reproduced observed free tropospheric GOM using
CMAQ (Bullock and Brehme, 2002) with GEM being oxidized primarily in gas-phase by O$_3$ and
OH, the latter being dominant, and found anticorrelation between GEM and GOM under the
dominance of photochemistry while positive correlation directly from emissions.  Lyman and
Jaffe (2011) found enhanced GOM concentrations of ~450 pg m$^{-3}$ and depleted GEM in one



stratospheric intrusion case, further speculated that the stratosphere was depleted in total Hg and
enriched in GOM, and suggested that stratospheric intrusion could be a source of GOM to the
troposphere.  Near Tullahoma, TN, USA the highest GOM concentrations (200 – 500 pg m$^{-3}$)
from flights over a year were observed always at 2 – 4.5 km altitude in 0 – 6 km vertical profiles
with a strong *seasonal variation* with a wintertime minimum and a summertime maximum
(Brooks et al., 2014).  In the same study, limited PBM measurements exhibited similar levels to
GOM at all altitudes.

In a most recent field campaign NOMADSS, the highest Hg(II) concentrations, 300–680

pg m$^{-3}$ were observed in dry (RH<35 %) and clean air masses during two flights over Texas at 5–
7 km altitude and off the North Carolina coast at 1–3 km (Gratz et al., 2015; Shah et al., 2016).
Gratz et al. (2015) found, using back trajectories, that a segment of air masses with elevated
GOM averaged at 0.266±0.038 ng m$^{-3}$ and ranging over 0.182 – 0.347 ng m$^{-3}$ at 7 km altitude
over Texas originated from the upper troposphere of the Pacific High.  It was speculated that the
stable, dry conditions of large scale anticyclones resulted in a lack of GOM removal by wet
deposition or in-cloud reduction and were thus ideal for GOM accumulation.   They
demonstrated that elevated BrOx could persist and that sufficient GOM could be produced
during long-range transport in the Pacific upper troposphere.  Their sensitivity analysis suggested
a range of 8 – 13 days required to produce the observed GOM.  Shah et al. (2016), using GEOS-
Chem with tripled bromine radical concentrations or a faster oxidation rate constant for GEM +
Br, simulated 1.5–2 times higher modeled Hg(II) concentrations and improved agreement with
the observations, and suggested that the subtropical anticyclones are significant global sources of
Hg(II).
**6. Summary and Recommendations**





This review summarized the general characteristics in GEM, GOM, and PBM

concentrations in the MBL, over land, from low to high latitudes, and from the surface to the
upper troposphere, and further the factors driving such variabilities based on a great wealth of
research in the literature. The Key points are summarized below.
1. For TGM/GEM in the MBL, diurnal variation mostly featured noon to afternoon minimums

due probably to in situ oxidation of GEM in the MBL in most oceanic regions, while in a few

studies, on TGM over the Atlantic and the equatorial Pacific Ocean, the opposite pattern was

observed with daytime maximums and attributed to enhanced oceanic evasion linked to

enhanced photoreduction and biological activity. Seasonal to annual variation was generally

characterized as higher (lower) concentrations in colder (warmer) months, which was largely

thought to be caused by less (more) loss via oxidation in colder (warmer) months. Long term

trends have been identified in three environments, Mace Head, Ireland, Canadian midlatitude

sites, and Cape Point, South Africa, and varied over different time periods, speculated to be

associated with changing anthropogenic and natural emissions as well as possibly redox

chemistry.

2. For MBL GOM, diurnal variation was generally characterized with noon to afternoon peaks

and nighttime low values, most likely driven by daytime GEM photooxidation involving

reactive halogens. Seasonal variation was often observed with higher concentrations in

spring and summer and lower in fall and winter, largely attributed to GEM photooxidation as

often supported by correlation of GOM with solar radiation and BrO. In one study

springtime maximums were also linked to biological activity and in a few studies annual

minimums were associated with scavenging by precipitation. No long term trends have been

reported for oceanic regions.



3. For MBL PBM, no consistent diurnal and seasonal variation has been identified in most studies, and only two studies reported seasonal variation with higher concentrations in fall/winter associated with anthropogenic emissions. Results from one study showed no consistent diurnal variation in Tekran measurements, but found a clear diurnal cycle with maximums at noon and minimums before sunrise using 10-stage impactor measurements.

4. For continental TGM/GEM, higher concentrations were found at urban sites than remote, rural, and elevated sites. This result is unbiased by elevated TGM/GEM from Asian sites. The predominant diurnal pattern was an early morning minimum and afternoon maximum, opposite to that at urban sites. Diurnal patterns at surface sites were thought to be driven by surface and local emissions, boundary layer dynamics, Hg photochemistry, dry deposition, and sequestering by dew. At elevated sites, mountain-valley winds appeared to be important drivers of the diurnal cycle. Seasonal variations were influenced by fossil fuel emissions for winter heating, surface emissions, and monsoons in Asia. At background sites, long-term declines in TGM are partially attributed to lower anthropogenic Hg emissions.

5. For continental GOM, concentrations were higher at elevated sites. However, this result may be biased by a large proportion of high elevation studies from China where speciated atmospheric mercury are typically elevated. The predominant diurnal pattern was a noon to mid-afternoon maximum and nighttime minimum, except for nighttime increases at urban and elevated sites. The driving mechanisms of the diurnal variations were suggested to include in situ photochemical production, dry deposition, and scavenging by dew. Entrainment of GOM from the free troposphere was believed to contribute to nighttime increases at some elevated sites. No predominant seasonal pattern in GOM was found, except for higher concentrations in the spring/summer at urban sites. Photochemical



production driven by strong solar radiation and atmospheric oxidants, free tropospheric
transport, anthropogenic emissions, and increased wet deposition during summer appeared to
be factors affecting the GOM seasonal pattern.
6. For continental PBM or TPM, no predominant diurnal pattern was found. Increase in PBM
or TPM was prevalent during colder seasons and are driven by local/regional coal
combustion and wood burning emissions, lower mixing height, reduced oxidation, and
increased gas-particle partitioning.
7. TGM/GEM over the ocean surface decreased from NH to SH with the highest concentrations
($\sim$3.5 ng m$^{-3}$) in northern hemispheric midlatitudes and the lowest in southern hemispheric
latitudes ($\sim$0.9 ng m$^{-3}$), as shown in the composite latitudinal distribution derive from studies
of the past three decades. This interhemispheric gradient was believed to suggest the
majority of Hg emissions in NH, refuting the hypothesis of large oceanic sources of Hg by
previous work. However, in other studies the largest oceanic source was found in the
equatorial region. Airborne measurement of TGM suggested distinct seasonal variation in
concentrations and latitudinal gradient, a $\sim$50 ppqv ($\sim$0.5 ng m$^{-3}$) increases in GEM
concentrations from $\sim$20°N – 30 °N to 60°N – 90°N latitudes in spring and negligible
latitudinal variation in summer. It was speculated that smaller latitudinal gradient of
temperature in summer likely enhanced meridional circulation resulting in smaller latitudinal
variation in GEM concentration in the troposphere.
8. Nearly constant, slightly decreasing GEM with altitude were shown in airborne
measurements in some regions, and depleted GEM was found in air masses under
stratospheric influence. Abundant GOM has been suggested, but only a handful of studies
have conducted measurements of GOM in the free troposphere showing concentrations of



hundreds of pg m$^{-3}$, particularly in the area of Pacific High.

There remain several outstanding unresolved questions and issues regarding our

understanding of the mechanisms controlling observed spatiotemporal variations in atmospheric
Hg, as listed follows.
1. Global distributions of tropospheric TGM/GEM, GOM, and PBM remain lacking despite

nearly two decades of extensive monitoring and modeling studies.  Speciated atmospheric

mercury in the continental boundary layer have been monitored in various regions of the

northern hemisphere, including in Asia, Europe and North America, and in different remote,

rural, urban, and high elevation environments; yet measurements remain scarce at inland

locations in the southern hemisphere.  In the MBL, most observations have been obtained via

shipboard measurements with a few exceptions as ground-based on islands, and subsequently

the global coverage was limited in space and time.  As a result, the diurnal variation to long-

term trends derived from such data suggested composite information instead of instantaneous

variation.  This limitation inevitably impedes the advancement of our understanding of the

factors controlling observed significant variation in atmospheric Hg concentrations.  In this

vein, it is therefore of paramount importance to have long-term monitoring of atmospheric

Hg continued in time and expanded in space, particularly over oceans perhaps utilizing

innovative platforms and at high altitudes, which certainly demands technological

breakthroughs in instrumentation.

2. GEM oxidation is one of the main driving mechanisms of the diurnal and seasonal variations

of TGM/GEM and GOM.  However, which oxidants are involved in the photochemical

reactions that could reproduce the diurnal and seasonal variations of GOM remain largely

unknown/uncertain, due to the lack of upper air measurements and speciated GOM



measurements, to a great extent a result of inadequate technologies, and a thorough
understanding of chemical reactions in atmospheric Hg transformation. Studies such as
Chand et al. (2008) estimated GOM concentrations using the reaction of GEM + OH alone,
and Sillman et al. (2007) reproduced observed GOM concentrations over Florida using
CMAQ with gas-phase oxidation of GEM by $O_3$ and OH only. However, the reactions of
GEM+ $O_3$ and GEM + OH have been subject to debate between theoretical and experimental
studies, as no mechanism consistent with thermochemistry has been proposed (Pal and Ariya,
2004; Calvert and Lindberg, 2005; Subir et al., 2011; Ariya et al., 2015). It was speculated
that GEM oxidation in the MBL and the upper troposphere was possibly largely Br-initiated
(Holmes et al., 2009; Gratz et al., 2015; Shah et al., 2016). This indicated that even if a
model reproduced observed concentrations of GOM, the chemistry in the model was not
necessarily correct. So far, most chemical transport models have largely focused on
reproducing annual and monthly variations in TGM/GEM (Lei et al., 2013; Song et al., 2015),
with large discrepancies between model simulations and surface measurements of GOM and
PBM (Zhang et al., 2012; Kos et al., 2012) . Future atmospheric modeling studies need to
focus more on reproducing the observed diurnal and seasonal variations using different Hg
photochemistry scenarios. Measurement studies need to include other oxidants besides
ozone (and BrO in limited number of studies) in the analysis of diurnal variation. Again, this
boils down to the most urgent need of fundamental understanding of the chemistry driving
atmospheric Hg cycling.
3. Mountain-valley atmospheric patterns appeared to be very common at elevated sites and
conducive to the entrainment of GOM from the free troposphere at night. Yet, the nighttime
increase in GOM did not seem to be widespread among elevated sites, with some sites



observing a GOM minimum at night. It implies spatial variability in GOM in the free
troposphere that warrants further study. Another related question is the degree of influence
the free troposphere has on surface GOM in the absence of katabatic winds. The use of
ozone and water vapor as tracers of the free troposphere may not be ideal considering
abundant ozone and water vapor near the surface.
4.  The higher summer TGM seasonal pattern was found to be more common among continental
sites impacted by surface emissions, whereas the seasonal TGM pattern characterized by
higher winter/spring concentrations tended to occur at sites affected by regional emissions
related to winter heating. Thus, it is important that studies on temporal variabilities broaden
their scope to include an analysis of source-receptor relationships.
5.  No definitive diurnal patterns in PBM measurements were found at MBL and continental
sites when measurements were collected using the Tekran speciation system. However,
impactor measurements in the MBL showed clearly-defined diurnal variation with daily
maximums at around noon and minimums before sunrise. PBM measurements need to
include particles of all size, as the current Tekran instruments could only measure PBM <2.5
μm.

**Acknowledgements**

The authors acknowledge the field technicians, students and/or researchers for collection

of speciated atmospheric mercury data that are summarized and discussed in this review paper.





**Table Caption**

Table 1: Summary of predominant temporal patterns of speciated atmospheric mercury at continental sites in the northern hemisphere

**Figure Captions**

Figure 1. Mean and ranges of TGM/GEM (a), GOM (b), and PBM (c) concentrations, estimated from the values in the literature as shown in Tables S1 – S3, over the Atlantic, Indian, Pacific, seas over the West Pacific (denoted as Pacific-Seas, only TGM/GEM in this category), seas in the Mediterranean region (denoted as Mediterranean), Arctic, and Antarctica Ocean. The solid black squares represent the mean value and the lowest whisker the minimum and the largest the maximum concentration in the region.

Figure 2. Median and range in TGM/GEM, GOM and PBM by site category (a) and by geographical region (b). Bar graph represents the median and error bar represents the maximum, estimated from the values in the literature as shown in Tables S4 – S6.

Figure 3. Compiled values for several marine/oceanic environmental systems. GEM over the Augusta basin is in red open circles. (Based on the figure from Bagnato et al., 2013)

Figure 4. GEM (ppqv) from the INTEX-B in spring 2006 and ARCTAS in spring and summer 2008 (Data sources: Talbot et al., 2007, 2008; Mao et al., 2010).





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

Table 1: Summary of predominant temporal patterns of speciated atmospheric mercury at
continental sites in the northern hemisphere

|  | Diurnal variation | Seasonal variation |
|---|---|---|
| *TGM/GEM* |  |  |
| Rural | Daytime maximum, nighttime minimum | Winter-spring maximum and summer-fall minimum |
| Urban | Nighttime maximum, daytime minimum | No predominant pattern |
| High elevation | Daytime maximum, nighttime minimum | Winter-spring maximum and summer-fall minimum |
| *GOM* |  |  |
| Rural | Midday to late afternoon maximum, nighttime minimum | No predominant pattern |
| Urban |  | Spring or summer maximum |
| High elevation | *Exception: nighttime maximum at urban and elevated sites | No predominant pattern |
| *PBM* |  |  |
| Rural | No predominant pattern | Maximum during heating season |
| Urban | No predominant pattern | Maximum during heating season *Exception: summer maximum |
| High elevation | No predominant pattern | Maximum during heating season |






a)

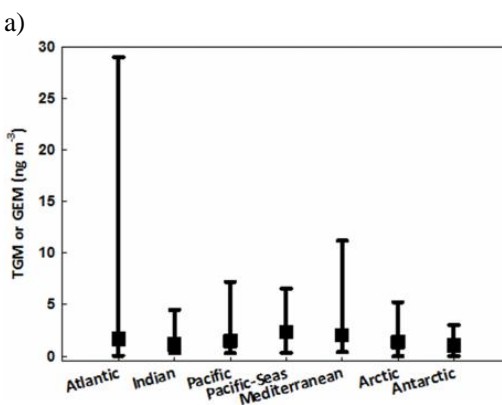

b)

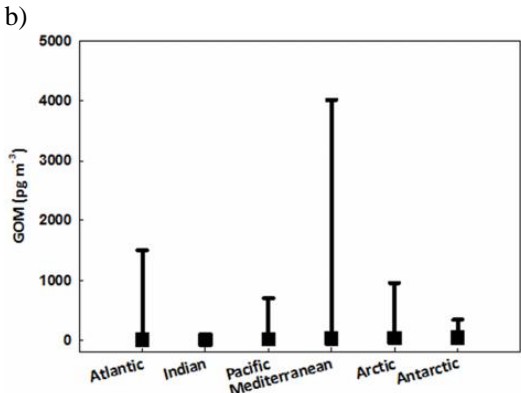

c)

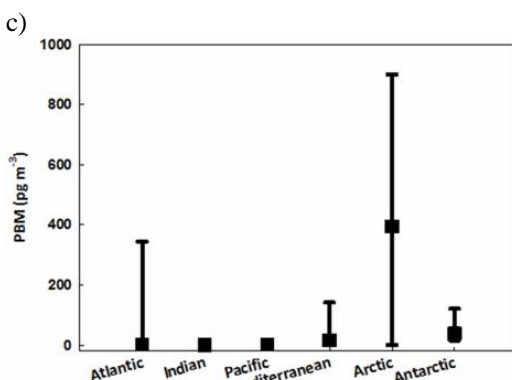


Figure 1. Mean and ranges of TGM/GEM (a), GOM (b), and PBM (c)
concentrations, estimated from the values in the literature as shown in Tables S1 –
S3, over the Atlantic, Indian, Pacific, seas over the West Pacific (denoted as
Pacific-Seas, only TGM/GEM in this category), seas in the Mediterranean region
(denoted as Mediterranean), Arctic, and Antarctica Ocean.  The solid black squares
represent the mean value and the lowest whisker the minimum and the largest the
maximum concentration in the region.






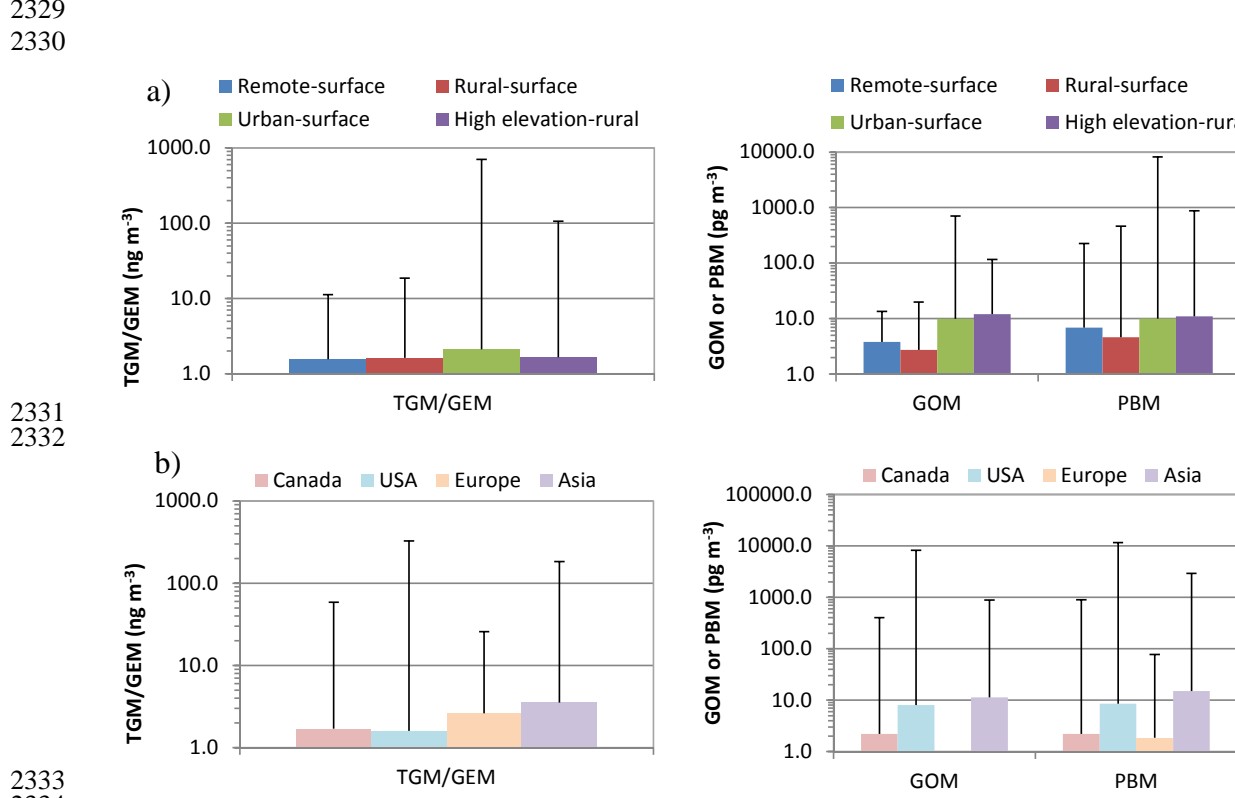


Figure 2. Median and range in TGM/GEM, GOM and PBM by site category (a) and by
geographical region (b). Bar graph represents the median and error bar represents the maximum,
estimated from the values in the literature as shown in Tables S4 – S6.






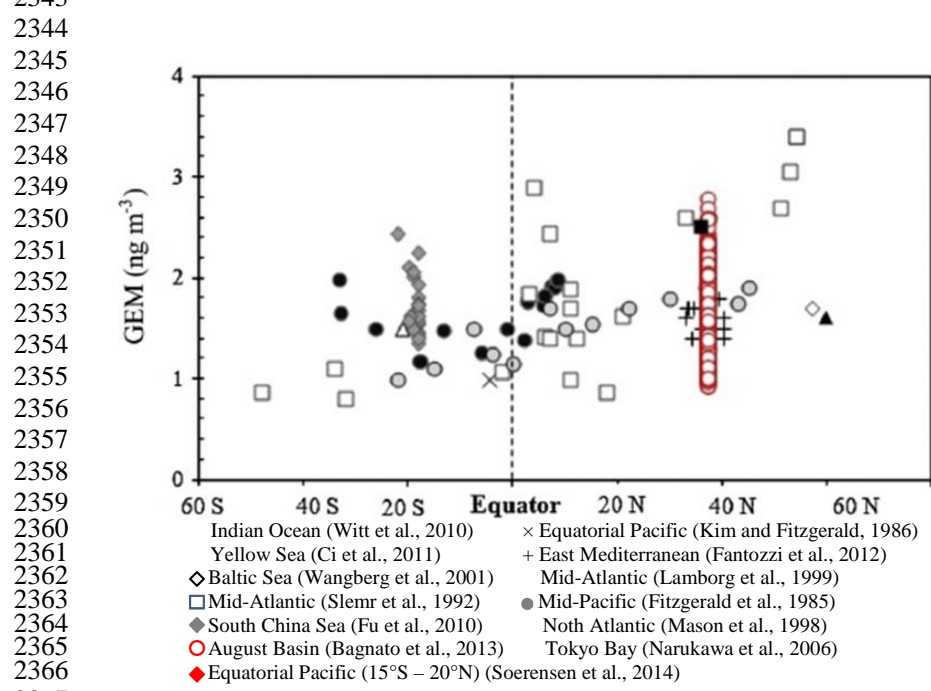

Figure 3. Compiled values for several marine/oceanic environmental systems. GEM over the Augusta basin is in red open circles. (Based on the figure from Bagnato et al., 2013)






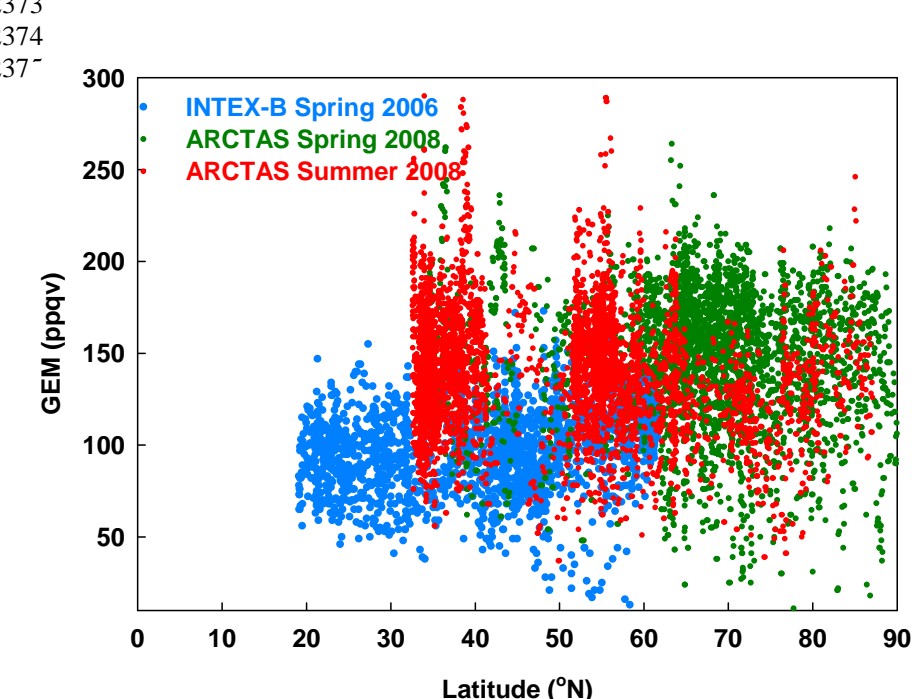

Figure 4. GEM (ppqv) from the INTEX-B in spring 2006 and ARCTAS
in spring and summer 2008 (Data sources: Talbot et al., 2007, 2008; Mao
et al., 2010).