# Peer review of "Current Understanding of the Driving Mechanisms for Spatiotemporal 1 2 3 Variations of Atmospheric Speciated Mercury: A Critical Review"

_Atmospheric Chemistry and Physics, 2016_

## Referee Comment (RC1) · Anonymous Referee #1 · 8 Aug 2016

07 August 2016

Review of H. Mao et al., Current Understanding of the Driving Mechanisms for Spatiotemporal Variations of Atmospheric Speciated Mercury: A Critical Review, *Atmospheric Chemistry & Chemistry Discussions*

**Summary**

Mao and colleagues present a review of published work on spatiotemporal patterns of atmospheric mercury. The authors have compiled an impressive volume of literature. I commend the authors on presenting an unbiased summary of published work. I recommend the review for publication after revision. Too much of the present manuscript feels like a reiteration of published work. The review could be greatly improved if it were more concise and provided a greater amount of critical insight.

**General Comments**

- The Abstract could use a statement motivating why we care about mercury in order to help appeal to a broader readership.  I also suggest tightening the conclusions and including at least one future research recommendation.
- The Introduction is unfocused and needs clearly stated objectives. Some of the content in the Introduction gets repeated in later sections. Delete redundancy wherever possible.
- The phrase "natural emissions" is used loosely and sometimes interchangeably with "re-emissions" or "legacy emissions". In light of the Minamata Convention, it is important to maintain clear language here and distinguish between natural primary sources (volcanism, outgassing of enriched mercuricferrous belts) and anthropogenic sources being remitted by land and ocean.
- Be concise. Delete unnecessary text. The current manuscript feels unnecessarily long.
- Old data (1960-80s) is included in comparisons alongside modern data -- is this really a valid comparison? At a minimum, it seems like it would be appropriate to comment on the major differences in analytical methods and the robustness of old data. I worry about the reliability of older data (Gustin et al., 2015).

**Specific Comments**

Line 62: Please include a citation for biodegradation. Biodegradation isn't a process commonly associated with atmospheric mercury.

Lines 101-106: Mao & Talbot (2012) is cited exclusively. Are there other references that could be included too?

Lines 536-549: Rivers and wastewater cannot explain North Atlantic trends in Soerensen et al. (2012) (Amos et al., 2014).

Line 527: The Pinatubo hypothesis is not widely embraced. I do not recommend including it in the review.

Line 811: Why would ship emissions be important? My understanding is most ships burning crude oil, which is low in Hg (Pironne et al., 2010).

Line 1468: "Refuting… large oceanic emissions". Please include a rationale for this conclusion. This is not an obvious conclusion from the review. If it's true, it's significant, but the conclusion needs to be buttressed with supporting evidence here in the Summary & Recommendations section.
Lines 1484-85: "Global distributions… remain lacking…" Delete? This statement is not particularly helpful.

Line 1492: "…trends derived from such data suggested composite information instead…" Perhaps there is a typo in the sentence? I'm unsure what the intended meaning is.

Figure 1: Break the y-axes, so you better see the variability in the data and the plots aren't dominated by one extra-large error bar.

---

## Referee Comment (RC2) · Anonymous Referee #2 · 10 Aug 2016

General Comments

This critical review presents a survey of a large body of literature regarding the spatiotemporal variations in speciated atmospheric mercury concentrations in a variety of environmental milieu (oceans, continents, high elevation, the free troposphere, and low to high latitudes). The authors are to be commended for pulling together such a body of work in an attempt to describe the current state-of-the-science as well as present current understanding, knowledge gaps and future necessary directions in this field. The manuscript is very long and at times repetitive and should be revised to make it more succinct.

Specific Comments Suggested revisions The summary and recommendations section

could be shortened considerably. Consider using bullet points especially for the "outstanding unresolved questions" section, e.g., this reviewer believes that point 1. could be condensed into – "Measurements in the southern hemisphere especially terrestrial locations are needed" while for point 2. Lines 1519-1520 capture the essence of what you are trying to say. Similarly points 3.,4. and 5 can be simplified with bullets.

This manuscript discusses work that spans decades. The authors have described published work along with literature interpretation. They should also provide their own interpretation of this body of work. Has the work led to greater understanding and if so why? With a view to the future should we continue using the same approaches?, the same measurement-based studies? Are innovative solutions needed to address the knowledge gaps delineated in the unresolved questions sections? If so, what are they?

Line 86 should be corrected to read "Statistics from studies prior to 2009 are referred to in Sprovieri et al. (2010b)"

Line 1029: is there a citation that can be used with this statement?

There is a tremendous amount of important data in the six tables in the supplementary information. This information could be made more appealing if presented on a plot showing latitude, longitude and concentrations.

---

## Author Response (AR1)

**Response to Reviewer #1**

**We greatly appreciate the reviewer's constructive comments, which have helped to improve the paper substantially.**

*Mao and colleagues present a review of published work on spatiotemporal patterns of atmospheric mercury. The authors have compiled an impressive volume of literature. I commend the authors on presenting an unbiased summary of published work. I recommend the review for publication after revision. Too much of the present manuscript feels like a reiteration of published work. The review could be greatly improved if it were more concise and provided a greater amount of critical insight.*

**R: We have improved the paper substantially through: (1) removing redundancy and unnecessary details; (2) summarizing common findings from multiple studies and pointing out differences between studies in each category/scenario, and (3) more importantly, providing more critical insights in the unresolved questions and recommendations for future research needs.**

*General Comments*
*The Abstract could use a statement motivating why we care about mercury in order to help appeal to a broader readership. I also suggest tightening the conclusions and including at least one future research recommendation.*

**R: The abstract was revised. In addition, per the reviewer's suggestions, we have added these statements in the abstract:**

**"Atmospheric mercury is a global pollutant and thought to be the main source for mercury in oceanic and remote terrestrial systems, where it becomes methylated and bioavailable, and hence atmospheric mercury pollution has global consequences for both human and ecosystem health."**

**"In examining the remaining questions and issues, recommendations for future research needs were provided, and among them again it boiled down to the most imminent need for GOM speciation measurements and fundamental understanding of multiphase redox kinetics."**

*The Introduction is unfocused and needs clearly stated objectives. Some of the content in the Introduction gets repeated in later sections. Delete redundancy wherever possible.*

**R: We have shorten the introduction by deleting most materials in the original 3$^{rd}$-6$^{th}$ paragraphs so as to avoid redundancy and keep it focused.**

*The phrase "natural emissions" is used loosely and sometimes interchangeably with "reemissions" or "legacy emissions". In light of the Minamata Convention, it is important to maintain clear language here and distinguish between natural primary sources (volcanism,*

*outgassing of enriched mercuricferrous belts) and anthropogenic sources being remitted by land*
*and ocean.*
**R: It is an important point. Corrections were made throughout the text.**
*Be concise. Delete unnecessary text. The current manuscript feels unnecessarily long.*
**R: See our response to the general comments above.**
*Old data (1960-80s) is included in comparisons alongside modern data -- is this really a valid*
*comparison? At a minimum, it seems like it would be appropriate to comment on the major*
*differences in analytical methods and the robustness of old data. I worry about the reliability of*
*older data (Gustin et al., 2015).*
**R: Point taken. The inclusion of old data was an attempt for the completeness of the review.**
**We agree the values were not comparable to those in more recent studies. Hence, the**
**comparison between the old and more recent data was mostly removed, and few retained**
**was revised with cautionary notes.**
*Specific Comments*
*Line 62: Please include a citation for biodegradation. Biodegradation isn't a process commonly*
*associated with atmospheric mercury.*
**R: This sentence is the continuation of the previous one and is followed by the next,**
**referring to mercury in the atmosphere and other spheres together in the Earth system.**
*Lines 101-106: Mao & Talbot (2012) is cited exclusively. Are there other references that could*
*be included too?*
**R: The introduction was condensed significantly to avoid redundancy. This reference was**
**removed in the introduction together with other material.**
*Lines 536-549: Rivers and wastewater cannot explain North Atlantic trends in Soerensen et al.*
*(2012) (Amos et al., 2014).*
**R: Amos et al. (2014) was added to counter the findings from Soerensen et al. (2012).**
*Line 527: The Pinatubo hypothesis is not widely embraced. I do not recommend including it in*
*the review.*
**R: The manuscript has been greatly revised and edited. This part has been removed in the**
**revised version.**

*Line 811: Why would ship emissions be important? My understanding is most ships burning*
*crude oil, which is low in Hg (Pironne et al., 2010).*
**R: Sprovieri et al. (2010) were making general statements, not exclusively with regard to**
**Hg, about ship emissions becoming a more important source of contaminants as emissions**
**from other sources were being more stringently controlled, and the Mediterranean was a**
**place where busy shipping routes ran close to population centers. The reference to**
**Sprovieri was revised to reflect this point and the reviewer's concern.**
*Line 1468: "Refuting… large oceanic emissions". Please include a rationale for this conclusion.*
*This is not an obvious conclusion from the review. If it's true, it's significant, but the conclusion*
*needs to be buttressed with supporting evidence here in the Summary & Recommendations*
*section.*
**R: This point was made by Temme et al. (2003a) based on the average NH/SH ratio of**
**TGM hemispheric median values and the higher variability in NH TGM concentrations**
**from their three cruises. Mason at al. (1994) hypothesized that oceanic emissions were a**
**large source to atmospheric Hg. Temme et al. (2003a) "refuted" this point by saying that**
**two thirds of oceans are located in the southern hemisphere and if oceanic emissions were**
**truly a large source, the large NH/SH ratio and large variability of TGM in the NH would**
**not have been likely. Temme et al. (2003a)'s cruises measurements covered the largest**
**areas in both hemispheres and were conducted along the same path three times and hence**
**cited. However, these are both hypotheses, and more studies suggested oceans as a source**
**in various oceanic regions. "Refute" is the word Temme et al. (2003a) used in their paper.**
**We changed "refuting" to "contradicting" now.**
*Lines 1484-85: "Global distributions… remain lacking…" Delete? This statement is not*
*particularly helpful.*
**R: We thought that this point is in fact quite important. For a compound such as ozone,**
**there have been numerous studies providing global distributions using satellite retrievals,**
**in situ measurements, and model simulations, so we have a fairly good idea of the global**
**distribution of ozone. In comparison, we do not really have such knowledge of GEM,**
**GOM, and PBM global distributions, despite decades of monitoring and modeling studies.**
**A lot of it remains controversial and speculative due to the lack of measurement data in the**
**southern hemisphere, in the marine boundary layer, and in upper air, and due to a lack of**
**the model simulations that we have confidence in.**
*Line 1492: "…trends derived from such data suggested composite information instead…"*
*Perhaps there is a typo in the sentence? I'm unsure what the intended meaning is.*
**R: We agree that the point did not come out right. The summary section was rewritten,**
**and the relevant point was reworded.**

*Figure 1: Break the y-axes, so you better see the variability in the data and the plots aren't*
*dominated by one extra-large error bar.*
**R: Done.**

 **Response to Reviewer #2**
**We greatly appreciate the reviewer's constructive comments, which have helped to**
**improve the paper substantially.**
*General Comments*
*This critical review presents a survey of a large body of literature regarding the spatiotemporal*
*variations in speciated atmospheric mercury concentrations in a variety of environmental milieu*
*(oceans, continents, high elevation, the free troposphere, and low to high latitudes). The authors*
*are to be commended for pulling together such a body of work in an attempt to describe the*
*current state-of-the-science as well as present current understanding, knowledge gaps and future*
*necessary directions in this field. The manuscript is very long and at times repetitive and should*
*be revised to make it more succinct.*
**R: We have improved the paper substantially through: (1) removing redundancy and**
**unnecessary details; (2) summarizing common findings from multiple studies and pointing**
**out differences between studies in each category/scenario, and (3) more importantly,**
**providing more critical insights in the unresolved questions and recommendations for**
**future research needs.**
*Specific Comments*
*Suggested revisions: The summary and recommendations section could be shortened*
*considerably. Consider using bullet points especially for the "outstanding unresolved questions"*
*section, e.g., this reviewer believes that point 1. Could be condensed into – "Measurements in*
*the southern hemisphere especially terrestrial locations are needed" while for point 2. Lines*
*1519-1520 capture the essence of what you are trying to say. Similarly points 3,4, and 5 can be*
*simplified with bullets.*
**R: The summary section was rewritten for the most part. Remaining questions were**
**discussed and recommendations were condensed into bullet points.**
*This manuscript discusses work that spans decades. The authors have described published work*
*along with literature interpretation. They should also provide their own interpretation of this*
*body of work. Has the work led to greater understanding and if so why? With a view to the future*
*should we continue using the same approaches? the same measurement-based studies? Are*
*innovative solutions needed to address the knowledge gaps delineated in the unresolved*
*questions sections? If so, what are they?*
**R: Per the reviewer's suggestions, discussions of such were added in the summary section.**
*Line 86 should be corrected to read "Statistics from studies prior to 2009 are referred to **in***
***Sprovieri et al. (2010b)"***
**R: Corrected.**

*Line 1029: is there a citation that can be used with this statement?*
**R: Four references were added, Conaway et al. (2005), Landis et al. (2007), Won et al.**
**(2007), and Pirrone et al. (2010).  The general view from Pirrone et al. (2010)'s review is**
**that the global contribution from petroleum fuels combustion represented 0.00013% of the**
**total anthropogenic emissions and thus can be neglected in global assessment.**
*There is a tremendous amount of important data in the six tables in the supplementary*
*information. This information could be made more appealing if presented on a plot showing*
*latitude, longitude and concentrations.*
**R: As the reviewer suggested, global maps of GEM, GOM, and PBM mean concentrations**
**at continental sites were plotted, as shown in Figure S1.  MBL concentration data usually**
**cover an extensive area or a long path, which we think is better represented in Figure 1**
**than could be in a global map.**

**A list of relevant changes made in the manuscript:**

1. Each one of the reviewers' comments and suggestions was addressed and corresponding changes were made throughout the entire manuscript.  Detailed changes please refer to the Response to Reviewers' Comments.

2. The manuscript has been edited and revised greatly based on the two reviewers' suggestions and comments.  It was shortened by 13 pages to remove redundancy and unnecessary details and make the key points and discussions more focused, as both reviewers suggested.

3. The title was changed to "Current Understanding of the Driving Mechanisms for Spatiotemporal Variations of Atmospheric Speciated Mercury: A Review", as our goals in this review were to be: 1) comprehensive and 2) objective.  Our interpretation of the current status of mercury research, discussions of remaining questions, and recommendations for future direction were built on the literature as a whole.

4. The abstract was revised to include motivation of mercury research to appeal to a broader audience, tighten the conclusions, and include one most urgent future research recommendation.

5. The summary section was largely rewritten with a list of key findings, remaining questions, and recommendations for future mercury research.

6.  Figures 1&2 were revised to enhance the presentation.

7. Global maps of GEM, GOM, and PBM for continental sites were provided as Figure S1 to present the important data in Tables S4 – S6, as Reviewer #2 suggested.

[revised manuscript text omitted]

Δ Indian Ocean (Witt et al., 2010)       × Equatorial Pacific (Kim and Fitzgerald, 1986)
○ Yellow Sea (Ci et al., 2011)          + East Mediterranean (Fantozzi et al., 2012)
◇ Baltic Sea (Wangberg et al., 2001)      ● Mid-Atlantic (Lamborg et al., 1999)
☐ Mid-Atlantic (Slemr et al., 1992)       ● Mid-Pacific (Fitzgerald et al., 1985)
◆ South China Sea (Fu et al., 2010)       ▲ Noth Atlantic (Mason et al., 1998)
○ August Basin (Bagnato et al., 2013)    ■ Tokyo Bay (Narukawa et al., 2006)
◆ Equatorial Pacific (15°S – 20°N) (Soerensen et al., 2014)

Figure 3. Compiled values for several marine/oceanic environmental systems adapted mostly from Bagnato et al. (2013)

[Figure]

Figure 4. GEM (ppqv) from the INTEX-B in spring 2006 and ARCTAS in spring and summer 2008 (Data sources: Talbot et al., 2007, 2008; Mao et al., 2010).